# Antioxidant, Antidiabetic, and Antiobesity Properties, TC7-Cell Cytotoxicity and Uptake of *Achyrocline satureioides* (Marcela) Conventional and High Pressure-Assisted Extracts

**DOI:** 10.3390/foods10040893

**Published:** 2021-04-19

**Authors:** Adriana Maite Fernández-Fernández, Eliane Dumay, Françoise Lazennec, Ignacio Migues, Horacio Heinzen, Patricia Lema, Tomás López-Pedemonte, Alejandra Medrano-Fernandez

**Affiliations:** 1Departamento de Ciencia y Tecnología de Alimentos, Facultad de Química, Universidad de la República, General Flores 2124, Montevideo 11800, Uruguay; afernandez@fq.edu.uy (A.M.F.-F.); tlopez@fq.edu.uy (T.L.-P.); 2Ingénierie des Agropolymères et Technologies Emergentes, Équipe de Biochimie et Technologie Alimentaires, Université de Montpellier, 2 Place Eugène Bataillon, 34095 Montpellier, France; eliane.dumay@orange.fr (E.D.); francoise.lazennec@umontpellier.fr (F.L.); 3Departamento de Química Orgánica, Facultad de Química, Universidad de la República, General Flores 2124, Montevideo 11800, Uruguay; imigues@fq.edu.uy (I.M.); heinzen@fq.edu.uy (H.H.); 4Grupo Tecnologías Aplicadas a la Ingeniería de Alimentos, Facultad de Ingeniería, Universidad de la República, Av Julio Herrera y Reissig 565, Montevideo 11300, Uruguay; plema@fing.edu.uy

**Keywords:** bioactive compounds, cell metabolism, flavonoids, high-hydrostatic pressure, marcela, phenolic compounds, TC7-cellular uptake

## Abstract

The growing incidence of non-communicable diseases makes the search for natural sources of bioactive compounds a priority for such disease prevention/control. *Achyrocline satureioides* (‘marcela’), a plant rich in polyphenols and native to Brazil, Uruguay, Paraguay, and Argentina, could be used for this purpose. Data on its antidiabetic/antiobesity properties and cellular uptake of bioactive compounds are lacking. The potentiality of non-thermal technologies such as high-hydrostatic pressure (HP) to enhance polyphenol extraction retains attention. Thus, in the present study aqueous and ethanolic marcela extracts with/without assisted-HP processing were chemically characterized and assessed for their in vitro antioxidant capacity, antidiabetic and antiobesity activities, as well as cellular cytotoxicity and uptake on intestinal cell monolayers (TC7-cells, a clone of Caco-2 cells). Aqueous and ethanolic conventional extracts presented different polyphenolic profiles characterized mainly by phenolic acids or flavonoids, respectively, as stated by reverse phase-high-performance liquid chromatography (RP-HPLC) analyses. In general, ethanolic extracts presented the strongest bioactive properties and HP had none or a negative effect on in vitro bioactivities comparing to conventional extracts. TC7-cell viability and cellular uptake demonstrated in conventional and HP-assisted extracts, highlighted the biological effects of marcela bioactive compounds on TC7-cell monolayers. TC7-cell studies showed no HP-induced cytotoxicity. In sum, marcela extracts have great potential as functional ingredients for the prevention/treatment of chronic diseases such as diabetes.

## 1. Introduction

*Achyrocline satureioides* (known by the popular name of ‘marcela’) could be used for the prevention/treatment of non-communicable chronic diseases including cardiovascular diseases, cancers, respiratory diseases, and diabetes [1], which are the main cause of deaths in the current times. Thus, the search for antioxidant, antidiabetic, and antiobesity natural sources is of great importance for their prevention/treatment. Marcela has been studied for its antioxidant, cell cytoprotective effect against oxidants [2], anti-inflammatory, analgesic, antispasmodic, constipating, sedative, immunomodulatory, antiviral, antiherpetic, choleretic and hepatoprotective actions, whereas partial cytotoxicity in mice and rats has been found for aqueous and ethanolic extracts [3]. It is a plant native to Brazil, Uruguay, Paraguay, and Argentina, commonly used as herbal tea [2,3,4]. Recently, marcela proved to present anti-cancer activity against glioma cell lines (U87, U251 and C6) and to be less cytotoxic to brain cell than gliomas [5]. However, no scientific studies regarding its antidiabetic and antiobesity underlying mechanisms have been reported. Marcela extracts are composed of flavonoids such as quercetin, luteolin and 3-*O*-methylquercetin in their glycosylated and aglycone forms [2], found in both ethanolic [6] and aqueous [2] extracts. These compounds possess several bioactive properties such as cytoprotective activity against oxidant agents [2], but there are no reports on the bioavailability and/or absorption experiments, neither about its cytotoxicity on intestinal cells (as a means to elucidate the effect after their ingestion), which are necessary to assess the potential effectiveness of the marcela bioactive compounds. Once absorbed, these compounds may exert the above-mentioned bioactivities.

Aqueous and ethanolic extracts have shown different polyphenolic profiles as a consequence of different polyphenols solubility correspondent to solvent polarity, with subsequently different bioactive properties and/or biological effectiveness. High hydrostatic pressure (HP) technology proved to increase polyphenolic extraction yields [7] and plant cell membrane damage [8]. HP can also disrupt weak bonds such as hydrophobic bonds subsequently generating conformational changes as well as denaturating cell proteins, which could lead to enhance compounds accessibility during extraction [8]. HP technology could be a resourceful procedure for *Achyrocline satureioides* polyphenols extraction by the use of moderate or no heat treatment [7,9], being especially useful for thermolabile compounds extraction [10]. However, these compounds could suffer modifications during the process. Thus, studies regarding bioactivity, absorption and cytotoxicity are needed in order to state if this innovative technology presents advantages related to conventional extraction procedures as well as to ensure this novel extracts are safe for consumption.

The aim of the present work is to evaluate *Achyrocline satureioides* antioxidant, antidiabetic and antiobesity properties of aqueous and ethanolic extracts compared to HP-assisted extracts, along with the exposure to cultures of intestinal cells in order to elucidate the degree of cytotoxicity (as assessed on cellular metabolic activity and cell membrane integrity) and uptake/absorption of extracted bioactive compounds.

## 2. Materials and Methods

### 2.1. Raw Material and Chemicals

*Achyrocline satureioides* (marcela) commercial samples were purchased in a pharmacy store (La Botica del Señor, Montevideo, Uruguay), and milled with a domestic coffee mill. All the reagents used in physicochemical characterization analyses were of reagent grade. Phenolic acids (gallic, chlorogenic and caffeic acids) and quercetin standards were purchased from Sigma-Aldrich (St. Louis, MO, USA) and used for marcela extract composition by reverse phase-high-performance liquid chromatography (RP-HPLC) and reverse phase-ultra-high-performance liquid chromatography (RP-UHPLC) analyses. Antioxidant assays reagents were purchased from Sigma-Aldrich (St. Louis, MO, USA): Folin reagent, 2,20-azinobis-(3-ethylbenzothiazoline-6-sulfonic acid) diammonium salt (ABTS), 6-hydroxy-2,5,7,8-tetramethylch-roman-2-acid (Trolox), fluorescein (FL) disodium salt, 2,20-azobis (2-methylpropionamidine) dihydrochloride (AAPH). Antidiabetic assays reagents were also purchased from Sigma-Aldrich (St. Louis, MO, USA): bovine serum albumin (BSA), methylglyoxal (MGO), aminoguanidine (AG), α-glucosidase (rat intestine acetone powder), acarbose, 4-methylumbelliferyl-α-D-glucopyranoside, human saliva α-amylase (Type IX-A), starch, maltose standard, 3,5-dinitrosalicylic acid. Antiobesity assay reagents were the following: lipase from porcine pancreas, 4-methylumbelliferyl oleate (4-MUO), and dimethyl sulfoxide (DMSO), which were purchased from Sigma-Aldrich (St. Louis, MO, USA), and orlistat standard was purchased from Alfa Aesar (Haverhill, MA, USA).

### 2.2. TC7-Cells and Reagents for Cell Culture

TC7-cells (a clone of Caco-2 cells) was kindly provided by Dr. Rousset (Centre de Recherche des Cordeliers, UMR S872, Paris, France). For cell culture, the following reagents were used.

High-glucose Dulbecco’s modified Eagle medium (DMEM) with L-glutamine and pyruvate (Phenol red-DMEM), high-glucose Dulbecco’s modified Eagle medium without L-glutamine and pyruvate (Phenol red-free DMEM), Dulbecco’s phosphate-buffered saline (DPBS) + Ca^2+^ and Mg^2+^, Hank’s Balanced Salt Solution (HBSS), penicillin-streptomycin mixture, MEM non-essential amino acid and foetal bovine serum (FBS) from Gibco^TM^ were purchased from Life Technologies (Villebon-sur-Yvette, France). For MTT-assay, 3-(4,5-dimethyl-2-thiazolyl)-2,5-diphenyl tetrazolium bromide (MTT) and dimethyl sulfoxide (DMSO) were purchased from Sigma-Aldrich (St-Quentin Fallavier, France). The β-nicotinamide adenine dinucleotide hydrate (NAD), Trizma^®^ base (Tris), l-(+)-lactic acid needed for LDH-assay, and Quercetin (96% dry matter, 95% purity) came from Sigma-Aldrich (St-Quentin Fallavier, France). Triton^®^ X-100 came from Merck (Darmstadt, Germany).

### 2.3. Methods

#### 2.3.1. Preparation of Marcela Extracts

Marcela aqueous extract (Mac) was obtained by adding 100 g of milled marcela powder to 1000 mL of distilled water. The mix was boiled for 1 h, filtered with paper (Whatman n°1) and the liquid was freeze-dried until constant weight (96 h).

Marcela ethanolic extract (Me) was obtained by adding 100 g of milled marcela powder to 1000 mL of ethanol (95%) followed by maceration at 20 °C for 24 h. The mixture was filtered with paper (Whatman n°1) then rotavaporated (60 °C, 120 rpm, under reduced pressure, approximately 13 kPa) to dryness and 30 mL of distilled water was added to recuperate polyphenols. Afterwards, the liquid was freeze-dried until constant weight (96 h).

High pressure (HP) extracts were prepared using a high-pressure unit Model S-IL-100-250-09W (HP Food Processor, Stansted Fluid Power, Ltd., Harlow, UK) located in Laboratorio Tecnológico del Uruguay pilot food plant (Montevideo, Uruguay). The pressure chamber (2 L volume, 100 mm bore internal diameter, 250 mm long) has inside the canister to hold samples. The vessel body and the pressure-transmitting fluid (water) were kept at treatment temperature (25 °C) by circulating water through an internal heat transfer jacket fitted to the outside of the high-pressure barrel assembly. The temperature of the pressure-transmitting fluid was monitored with a thermocouple positioned at the chamber bottom. Before treatment, samples were individually packed in Cryovac^®^ pouches (Sealed Air^®^, Charlotte, North Carolina, USA) by adding 10 g of milled marcela powder to 100 mL of phosphate buffer pH 7.9 for aqueous HP extract, or to 100 mL of ethanol (95%) for ethanolic HP extract, then vacuum sealed. Samples were introduced in the pressurization chamber previously thermostated at 25 °C then submitted to 400 MPa and 25 °C for 1 min, in the case of ethanolic HP extract (Me HP), or 200 MPa and 25 °C, at pH 7.9 for 1 min for aqueous HP extract (Mac HP). These conditions were selected in previous trials studying optimum conditions for marcela antioxidant compounds extraction through HP procedure. Pressure was raised from 0.1 MPa at a rate of 100 MPa per 30 s, maintained at the desired pressure level for 1 min then reduced down to 0.1 MPa in less than 30 s. Sample blanks were also prepared in the same way at 0.1 MPa and 25 °C for 1 min but without the pressure-processing step, by adding 10 g of milled marcela powder to 100 mL of phosphate buffer pH 7.9 for aqueous HP blank extract (Mac HP BL), or to 100 mL of ethanol (95%) for ethanolic HP blank extract (Me HP BL). Afterwards, liquid samples were freeze-dried until constant weight (96 h). All the extracts were stored at −20 °C for further analyses.

#### 2.3.2. Proximate Analysis

To characterize the initial sample or raw material (marcela powder), different parameters were determined: fat, protein, ashes, dietary fiber, moisture, and total carbohydrates (by difference using protein, moisture, ashes, and fat content). All determinations were performed at least in triplicate as in Association of Official Analytical Chemists (AOAC) [11] methods. Briefly, protein content was determined by Kjeldhal method using the conversion factor 6.25, moisture was determined using a conventional oven at 105 °C till constant weight, ashes was determined by using a furnace at 525 °C for 8 h, and fat content was obtained by Soxhlet method for 6 h using petroleum ether.

#### 2.3.3. RP-HPLC and RP-UHPLC Analyses

To obtain the polyphenolic profile of extracts, each of the extract samples were eluted by RP-HPLC (Shimadzu, SPD-20A detector and LC-10AT pump) according to De Souza et al. [6] with detection at 370 nm in a Jupiter C18 reverse phase column and an isocratic flow program of 1 mL/min. The mobile phase was composed by methanol:0.16 M phosphoric acid, in a ratio of 53:47 (*v*/*v*). The injection volume was 20 μL.

Each of the samples were also eluted by RP-UHPLC according to Reza et al. [12] with some modifications. RP-UHPLC UltiMate 3000 (Thermo Fisher Scientific, Massachusetts, USA) was used with a diode array detection (DAD detector). The reverse phase column was a Thermo Scientific BDS Hypersil C18 (150 × 3 mm, 3 µm particle size) used at 1 mL/min flow rate. Mobile phase was composed by methanol, phosphoric acid (pH 2.81) and acetonitrile in gradient: time 0 min, 5% acetonitrile and 95% phosphoric acid (initial condition); time 10 min, 10% acetonitrile, 10% methanol and 80% phosphoric acid; time 20 min, 20% acetonitrile, 20% methanol and 60% phosphoric acid; time 40 min, 20% acetonitrile, 20% methanol and 60% phosphoric acid; time 45 min, 100% acetonitrile; time 50 min, 100% acetonitrile; time 55 min, 5% acetonitrile and 95% phosphoric acid (to return to the initial conditions). The duration of each run was 55 min. The injection volume was 20 μL. The software used was Dionex Chromeleon 7.1 SR2. Phenolic acids were quantified by detection at 290 nm and quercetin was quantified at 370 nm. Phenolic acids and quercetin were identified and quantified by the use of pure standards and the construction of calibration curves through the detection at 290 and 370 nm for phenolic acids and quercetin, respectively.

#### 2.3.4. Antioxidant Capacity

Total polyphenol content was performed by Folin–Ciocalteau method [13] as described by Fernández-Fernández, Iriondo-DeHond, Dellacassa, Medrano-Fernandez, and del Castillo [14], preparing sample solutions in distilled water and using a gallic acid standard curve (0.05–1.0 mg/mL). Results were expressed as mg GAE/g extract.

The ABTS method [15] was performed as described by Fernández-Fernández et al. [14], preparing samples in phosphate buffer (pH 7.4) and using a Trolox calibration curve (0.25–1.5 mM). Results were expressed as µmol TE/mg extract.

Oxygen radical antioxidant capacity-fluorescein (ORAC-FL) assay was performed by the method of Ou, Hampsch-Woodill, and Prior [16] modified by Dávalos, Bartolomé, and Gómez-Cordovés [17] as described in Fernández-Fernández et al. [14]. The area under the curve (AUC) of fluorescence vs. time were calculated and normalized to the AUC of the blank as follows: AUC_antioxidant (trolox or sample)_-AUC_blank_. Trolox calibration curve (AUC vs. [Trolox]) was constructed and results were expressed as μmol TE/mg extract.

All samples were prepared in triplicate and each one of the preparations was measured in triplicate.

#### 2.3.5. Antidiabetic and Antiobesity Activities

α-glucosidase inhibition capacity was evaluated as described by Fernández-Fernández et al. [14] as an antidiabetic strategy. Briefly, fluorescence measurements were displayed at 37 °C for 30 min (each minute) at 360 and 460 nm of excitation and emission wave lengths, respectively. Acarbose was used as reference with probed inhibition capacity.

α-amylase inhibition assay was evaluated as another antidiabetic strategy and performed as reported by Li, Yao, Du, Deng, and Li [18] with some modifications described by Fernández-Fernández et al. [19]. The inhibition capacity was calculated by taking positive control as 100% of enzyme activity.

Fluorescent advanced glycation end products (AGEs) formation was evaluated by determining BSA-MGO formation inhibition (antiglycant capacity) as another antidiabetic strategy, obtaining the IC_50_ value [20]. Briefly, sample mixtures consisted of 500 µL BSA stock solution (2 mg/mL in PBS, 1 mg/mL final concentration), 25/50 µL 5/10 mM MGO stock solution (200 mM in PBS, 5 mM final concentration), different volumes of extracts from a sample stock solution of 50 mg/mL (concentrations 0.1‒5 mg/mL) plus sufficient volume of PBS 10 mM pH 7.4 with 0.02% sodium azide to achieve 1 mL of the mixture final volume. Sample blanks consisted of the samples (different concentrations) with sufficient volume of PBS to achieve 1 mL of the mixture final volume (intrinsic fluorescence of the samples). Positive control was prepared by mixing 500 µL BSA, 25/50 µL 5/10mM MGO and 475/450 µL PBS, as previously explained. Aminoguanidine (AG) was used as reference (1, 4 and 8 mM final concentrations) mixed with BSA, MGO and PBS. All stock solutions were prepared in PBS 10 mM pH 7.4 with 0.02% sodium azide. Eppendorf tubes were incubated at 37 °C for 7 days. Fluorescence measurements were performed at 340 and 420 nm of excitation and emission wavelengths, respectively, and inhibition percentages were calculated by taking positive control as 100% of AGEs formation.

Pancreatic lipase inhibition capacity was determined as described in Fernández-Fernández et al. [14]. Measurements were determined after 30 min incubation at 25 °C by fluorescence measurements at 360 nm and 460 nm of excitation and emission wavelengths, respectively.

#### 2.3.6. TC7-Cell Culture and Sample Deposits

TC7-cells were routinely grown according to Benzaria et al. [21,22] with minor changes in 75 cm^2^ sterile cell culture flasks in phenol red-DMEM culture medium. TC7-cells (passages 41‒49) were seeded in sterile 12-well plates (3.5 cm^2^/well; Nunc, VWR, Fontenay-sous-Bois, France) at a density of 2.5 × 10^5^ cells/well (1 mL of cell suspension/well) then cultivated at 37 °C in controlled atmosphere (8% CO_2_, 92% air, 100% relative humidity, RH) (Thermo Scientific 8000 incubator, Thermo Electron, St-Herblain, France) for 19–20 days to reach cell-confluence and obtain differentiated cells, the culture medium (phenol red-DMEM supplemented with 20% *v/v* heat-inactivated FBS, 1% *v*/*v* penicillin-streptomycin and 1% *v/v* MEM non-essential amino acids), being changed every 2 days. Cell confluence was assessed by transepithelial electrical resistance (TEER; Millicell^®^-ERS volt-ohm meter, Millipore, St-Quentin-en-Yvelines, France) measurements before deposing extract samples onto the cells. For TEER measurements, cells were grown in sterile Transwell plates with ThinCert inserts (3 μm pore size; 1.13 cm^2^/well; Greiner Bio-one, VWR International, Fontenay-sous-Bois, France) at a density of 2.5 × 10^5^ cells/well, obtaining TEER values of 750–800 Ω cm^−2^. Cell confluency of the cell cultures was also checked by inverse phase microscope examination.

After washing using Phenol red-free DMEM, differentiated TC7-cells were incubated for 3 h or 22 h at 37 °C in controlled atmosphere (8% CO_2_, 92% air, 100% HR) with 500 µL of extract mixture or control sample. Exposure times (3 h or 22 h) were chosen on the basis of previous experiments [22], and taking into account the open time necessary to prepare cell series. All cell seeding and sample deposit experiments were carried in sterile conditions under a laminar flow cabinet (PSM MSC Advantage, ThermoFisher Scientific, St-Herblain, France), using 0.2 µm filtrated media, sterile solutions and sterile plastic material (pipets, tips, flasks, plates, microplates, Eppendorf^®^ and Falcon^®^ tubes).

Ethanolic and aqueous extracts were assayed on TC7-cells for a range of lyophilized extract concentrations in the cell deposit medium. For this, a 56.6 mg of lyophilized ethanolic extract (Me) was solubilized into 1 mL of 80% Ethanol (Ethanol/Water 80:20, *v*/*v*), and 28.3 mg of lyophilized aqueous extract (Mac) into 1 mL of sterile distilled water, due to its solubility. Various concentrations of lyophilized extracts were then prepared ranging from 0.88 to 56.6 mg/mL in 80% Ethanol for Me, and ranging from 3.54 to 28.3 mg/mL in sterile distilled water for Mac. In the case of HP extracts, samples were solubilized in the range 0.88–14.2 mg/mL in 80% Ethanol for Me HP, and in the range 3.54–28.3 mg/mL in sterile distilled water for Mac HP. Mixtures of 100 μL of the latter extract solutions and 1.9 mL of Phenol red-free DMEM were then prepared for cell deposit in 12-well plates. A 500 μL of mixture per well was deposed onto TC7-cells in triplicate. Control samples were also prepared using Phenol red-free DMEM alone, or 1.9 mL Phenol red-free DMEM plus 100 µL of one of the following solutions: distilled water, 80% Ethanol, 0.1% Triton X100, or purified quercetin at 0.156–0.625 mg/mL in 80% Ethanol.

#### 2.3.7. Determination of In Vitro TC7-Cell Membrane Integrity and Cell Metabolic Activity

After 3 h or 22 h of exposure time with Me or Mac extracts, or control samples, the apical TC7-culture supernatants were collected on ice to determine the lactate dehydrogenase (LDH) activity, i.e., LDH-release from cytosols in cellular apical media. TC7-cells were recovered for the MTT colorimetric determination (i.e., evaluation of cellular metabolic activity or cell viability), both as described by Benzaria et al. [22] with minor modifications. LDH-leakage outside TC7-cells was determined to evaluate cellular membrane damage after exposure to extracts, as an indicative of further cell death. Apical TC7 media were collected then four-fold diluted (1/4) in Phenol red-free DMEM. A 25 µL of the latter solutions were added to 96-well plates (8 replicates for each apical diluted medium). Then, 250 µL of pH 9.3 NAD reagent (1.65 mM NAD, 165 mM KCl, 54 mM L-lactic acid, 108 mM Tris, final) was added per well. LDH induced the lactate oxidation into pyruvate with the simultaneous reduction of NAD to NADH. NADH absorbance was therefore measured in plate wells at 340 nm and 37 ˚C over 10 min (Multiskan Spectrum microplate reader, Thermo Electron, Vintaa, Finland). LDH activity was expressed as the difference in absorbance values taken at 0 and 10 min. Results were the means of eight absorbance determinations for each apical cell medium. A positive control was included in the series, corresponding to high LDH release by exposure of TC7-cells to Triton^®^ X-100 in Phenol red-free DMEM (0.005% final, *v*/*v*).

TC7-cells in plate wells were then recovered for MTT-assay to evaluate cell viability after 3 h or 22 h of exposure time to Me or Mac extracts at various concentrations, or to control samples. After washing with Phenol red-free DMEM, cells were incubated for 3 h with 500 µL/well of MTT reagent (0.15 mg/mL MTT in FBS-free Phenol red-free DMEM) at 37 °C. MTT is reduced into Formazan^®^ by a succinate dehydrogenase in living cells. After removing MTT solution, Formazan^®^ was recovered by cell-lysing for 30 min at 37 °C using 1000 µL DMSO per well. Amounts of 100 µL were then transferred into 96-well plates to measure Formazan^®^ absorbance at 570 nm (Multiskan Spectrum microplate reader) (8 replicates for each apical cell lysate). The cell ability to reduce MTT provides an indication of mitochondrial integrity, and therefore of cell metabolic activity or cell viability. Results were expressed as the means of 8 absorbance determinations for each apical cell lysate sample.

For each series (“3 h or 22 h of exposure time”), data were pooled from 4 independent cell culture experiments involving different TC7-cell passages. For Me and Mac extracts, 3 to 4 independent cell culture experiments were carried out on different days, and 1 to 3 for Me HP and Mac HP, each with currently 2-3 apical cell media analyzed per studied extract concentration.

#### 2.3.8. Marcela Bioactive Compounds’ Uptake

Cellular uptake of marcela compounds was determined after TC7-cell exposure to Me or Mac extracts for 3 h or 22 h, in triplicate, as described in Section 2.3.6. Me and Me HP samples were deposed at 0.088–0.354 mg/mL extract onto TC7-cells; Mac and Mac HP samples, at 0.71–1.42 mg/mL extract. After cell incubation, the apical culture media were taken off and the plate-wells washed with DPBS+. TC7-cells were scratched with 500 µL of cold acidified methanol (methanol with 0.1% *v/v* acetic acid) and transferred into Eppendorf tubes for centrifugation at 10,000 rpm and 4 °C for 5 min. Methanolic supernatants of centrifugation were kept in brown vials at 4 °C until further analysis by RP-UHPLC as described in Section 2.3.3. Prior to sample injection (20 μL) in RP-UHPLC, supernatants were dried at 40 °C in a conventional stove and re-suspended in 250 µL of methanol.

#### 2.3.9. Statistical Analysis

All experiments were performed in triplicate and cell studies were performed at least in three different passages. The statistical analysis was established by analysis of variance (ANOVA). Results were expressed as means ± standard deviation (SD) (*n* = 3). Tukey test was applied to determine significant differences between values (*p* < 0.05) using Infostat v. 2015 program. Different letters state significant differences when *p* < 0.05.

In the case of cell studies, the statistical analysis was carried out using Sigma Plot vs. 11.0 program: the pooled data were analyzed by one-way analysis of variance (ANOVA) with all pairwise multiple comparison procedure (Holm–Sidak test) and an overall significance level of 0.05.

## 3. Results and Discussion

### 3.1. Chemical Composition

Aqueous and ethanolic extracts resulted in a yield of 6.9 and 3.1% *w*/*w*, respectively. The results of the proximate analysis showed in marcela powder (raw material) high fiber and carbohydrates contents. Marcela powder contained per 100 g: 4.77 ± 0.02 g proteins, 4.52 ± 0.18 g lipids, 21.01 ± 0.79 g carbohydrates (without fiber), 57.22 ± 0.73 g fiber, 4.94 ± 0.05 g ashes, and 7.54 ± 0.13 g moisture.

As to RP-HPLC (Figure 1A,B), and RP-UHPLC results (Figure 1C,D), the extracts presented a typical chromatogram as previously reported by De Souza et al. [6] in *Achyrocline satureioides* preparations. According to De Souza et al. [6] the three prominent peaks correspond to quercetin, luteolin and 3-*O*-methylquercetin, in order of appearance in the chromatogram (Figure 1B). In the present study, retention times (RT) were lower because of using a higher flow (1 mL/min) than 0.6 mL/min. Furthermore, the last prominent peak at RT 19 min in Me chromatogram (Figure 1B) could correspond to achyrobichalcone according to Zorzi et al. [23]. Polyphenol profile of marcela extracts in the current study are also in agreement with those reported by Martínez-Busi et al. [24].

Chromatograms at 370 nm showed that Me extract presents mainly flavonoids such as 3-*O*-methylquercetin (30% in the extract as stated in relative area, RA), being in a close proportion to quercetin (22% RA) (Figure 1B). In contrast, Mac chromatogram at 290 nm presented mainly phenolic acids (over 65% RA, Figure 1C) compared to flavonoids in which quercetin (5% RA) is in lower proportion than 3-*O*-methylquercetin (7–8% RA) as shown by Mac chromatogram at 370 nm (Figure 1A). These results agree with those reported by Polydoro et al. [25] where the aqueous extract of *Achyrocline satureioides* presented the lowest contents of quercetin, luteolin and 3-*O*-methylquercetin. Moreover, they found higher concentrations of the latter compounds in the extract with higher proportion of ethanol (80%) with similar ratio of quercetin to 3-*O*-methylquercetin. In the present study, quercetin was quantified in Mac and Me extracts by RP-HPLC obtaining 1.98 ± 0.13 and 88.9 ± 6.36 mg of quercetin/g extract, respectively. Calculating from quercetin calibration curve, 3-*O*-methylquercetin was estimated to 3.3 ± 0.3 mg and 127.1 ± 9.1 mg/g extract for Mac and Me, respectively. The Me extract in the present study showed greater quercetin content than the marcela aqueous extracts prepared by maceration and ultrasound-assisted extraction reported by Guss et al. [26].

In parallel, Me and Mac extracts were eluted by RP-UHPLC method (Figure 1C,D) in which, phenolic acids were identified and quantified. The composition in phenolic compounds is associated with the type of solvent used. Mac was characterized by the presence of gallic acid (RT 1.1 min), chlorogenic acid (RT 4.6 min) and caffeic acid (RT 5.4 min) with 11.9, 15.1 and 18.7% RA, respectively (Figure 1C) which corresponds to 3.76 ± 0.25 mg of gallic acid, 14.28 ± 0.01 mg of chlorogenic acid and 3.28 ± 0.62 mg of caffeic acid per g of Mac extract, in agreement with previous reports [27,28].

### 3.2. Antioxidant Capacity

As to total polyphenol content, aqueous extracts (Mac, Mac HP BL and Mac HP) presented lower content than ethanolic ones (Me, Me HP BL and Me HP) indicating ethanol favors polyphenols extraction (Table 1). For ABTS antioxidant capacity (Table 1), the tendency was different resulting Me as the best, followed by all the other extracts with no significant differences (*p* > 0.05). For ORAC-FL antioxidant capacity (Table 1), the highest antioxidant potential were Mac and Me, followed by ethanolic extracts Me HP BL and Me HP. When analyzing HP-assisted extracts, both aqueous and ethanolic HP extracts (Mac HP and Me HP) presented a lower polyphenol content by 3.2% and 7.6%, respectively (although non-significant, Table 1) when compared to their respective blanks (Mac HP BL and Me HP BL). Thus, high hydrostatic pressure did not significantly affect total polyphenol content nor antioxidant capacity when applied at the tested conditions to the crude extracts dispersed in phosphate buffer or 95% ethanol (Section 2.3.1).

Compared to other studies, these extracts presented similar total polyphenol content to the marcela extracts reported by Ferraro et al. [29] (23.0‒112.6 mg GAE/g) with the highest polyphenol content, observing a correlation with antioxidant capacity determined by DPPH. In contrast, Guss et al. [26] reported greater polyphenol content and ABTS antioxidant capacity (i.e., lower IC_50_ value) of marcela maceration and ultrasound-assisted ethanolic extracts. In the current work, Me IC_50_ value was of 354 ± 25 µg/mL compared to 21.8 ± 0.8 and 21.3 ± 0.4 µg/mL for marcela maceration and ultrasound-assisted ethanolic extracts [26]. Marcela ethanolic extract (Me) presented higher total polyphenol content when compared to other medicinal herbs such as *Mentha x piperita* L., *Peumus boldus* Mol. and *Baccharis trimera* Iless. aqueous and ethanolic extracts [30], as well as antioxidant capacity. In contrast with Irazusta et al. [30] results, marcela aqueous extract showed lower antioxidant capacity than ethanolic extract. Antioxidant capacity of crude methanolic extracts of native Australian mint and common spearmint showed 398.5 ± 19.3 and 403.5 ± 14.8 µmol TE/g extract for ABTS, and 1727.2 ± 183.5 and 1551.1 ± 137.4 µmol TE/g extract for ORAC-FL, respectively [31], presenting lower antioxidant capacity than Mac and Me extracts in the present study (Table 1).

### 3.3. Antidiabetic Activities

α-amylase and α-glucosidase inhibition capacities were assessed (Figure 2A,B) as a strategy for post-prandial plasma glucose level regulation through delay/inhibition of complex carbohydrates hydrolysis during digestion, such as starch, causing lower glucose absorption [14]. For α-amylase inhibition (Figure 2A), acarbose and quercetin presented the highest inhibitions with IC_50_ values of 34.1 ± 0.8 and 2.4 ± 0.2 µg/mL, respectively. As to the extracts, aqueous extracts presented very low inhibition at the tested concentrations (up to 25 mg/mL, data not shown), in contrast with ethanolic extracts which showed IC_50_ values of 515 ± 44 (Me), 2900 ± 51 (Me HP BL) and 7974 ± 422 µg/mL (Me HP), demonstrating HP negatively affects α-amylase inhibition capacity. Moreover, quercetin seems to be one of the responsible for ethanolic extracts inhibition capacity, because of being one of the main compounds present in the latter extracts.

For α-glucosidase inhibition (Figure 2B), acarbose (IC_50_ = 4.0 ± 0.3 µg/mL) and chlorogenic acid (IC_50_ = 69.1 ± 1.6 µg/mL) presented the highest inhibition capacity (lowest IC_50_ value). Mac extract presented an IC_50_ value of 150.8 ± 54.0 µg/mL, and 157.6 ± 23.3 µg/mL was found for Me extract with no significant differences (*p* > 0.05). For Mac HP and its blank (Mac HP BL), IC_50_ values were 2973.1 ± 403.2 and 5392.0 ± 437.1 µg/mL respectively (significant difference for *p* < 0.05), stating bioactive compound extraction was favored by high hydrostatic pressure. In the case of Me HP and its blank (Me HP BL), IC_50_ values were 2587.3 ± 214.5 and 2211.1 ± 196.0 µg/mL, respectively, negatively affecting bioactivity by HP but with no significant differences (*p* > 0.05).

In accordance with the present work, quercetin has shown to possess more α-amylase inhibition capacity than acarbose [32]. Furthermore, other extracts from medicinal plants/herbs (Vietnamese and Amazonian plants, *Agrimonia asiatica*, species of *Myrcia* genus and *Euphorbia hirta* herbs) possessing phenolic acids (e.g., gallic acid) and flavonoids (quercetin and/or quercetin derivatives, and luteolin) like marcela extracts, have shown antidiabetic properties (α-amylase and α-glucosidase inhibition capacity) [33,34,35,36]. Particularly, *Euphorbia hirta* L. extract has shown to lower fast blood glucose level after 4 h and a significant reduction after 15 days treatment in streptozotocin-diabetic mice [36]. Guava (*Psidium guajava* L.) leaves possessing gallic, caffeic and chlorogenic acids, and quercetin, among others, have also shown antidiabetic properties [37]. The latter reports show the potential that marcela extracts could have as functional ingredients.

As another strategy for diabetes complications prevention/treatment, there is the inhibition of AGEs formation. Figure 2C showed maximum inhibition (close to 100%) of AGEs formation for Me extract at all the concentrations tested (0.1–5 mg/mL) in contrast with Mac extract that presented an increased inhibition trend with increasing concentration, although not significant (*p* > 0.05). This indicates that Me presents higher antiglycant capacity than Mac. Moreover, extracts did not present any differences when compared to methylglyoxal (MGO) at 5 and 10 mM. Inhibition capacity was not affected by MGO concentration at the tested concentrations (5 and 10 mM). Other medicinal herbs used as infusions, like marcela, have shown to inhibit AGEs formation such as *Mentha x piperita* L., *Peumus boldus Mol.* and *Baccharis trimera Iless.* [30] in a similar level as marcela extracts. Ethanolic extracts of ten common household condiments/herbs (*Allium sativum*, *Zingiber officinale*, *Thymus vulgaris*, *Petroselinum crispum*, *Murraya koenigii Spreng*, *Mentha piperita* L., *Curcuma longa* L., *Allium cepa* L., *Allium fistulosum* and *Coriandrum sativum* L.) showed correlation between total polyphenol content, antioxidant capacity and anti-glycant capacity [38], showing the same tendency when comparing marcela aqueous and ethanolic conventional extracts. Ethanolic extract (Me) showed higher total polyphenol content, antioxidant and anti-glycant capacity than aqueous extract (Mac).

### 3.4. Antiobesity Activity

Lipase inhibition capacity was determined (Figure 2D) as a strategy for post-prandial fat absorption control during digestion by delay/inhibition of triglycerides hydrolysis into free fatty acids, leading to lower fat absorption [14]. Mac extract presented an IC_50_ value of 1.471 ± 0.103 mg/mL and ethanolic extract (Me) 0.219 ± 0.028 mg/mL, the latter having no significant difference (*p* > 0.05) with Orlistat IC_50_ value (1.9 ± 0.2 µg/mL). Mac HP extract presented very little inhibition at the tested concentrations (0.1–10 mg/mL) and lower for its blank without pressure (Mac HP BL) (data not shown). Me HP extract presented an IC_50_ value of 2.025 ± 0.053 mg/mL and its blank without HP (Me HP BL) of 1.634 ± 0.038 mg/mL, the latter having no significant differences with Mac (*p* > 0.05). Considering all extracts, ethanolic extracts presented the best inhibition capacity, although HP negatively affected the inhibition capacity when compared to the blank (increased IC_50_ value of Me HP compared to Me HP BL) with significant differences (*p* < 0.05). Aqueous extracts seems to be more bioactive with applied high temperature (boiling extraction, Section 2.3.1) and for ethanolic extracts, it seems as if time was a key factor for bioactive compounds solvent extraction.

In parallel, polyphenol standards were tested finding IC_50_ values of 4.566 ± 0.231, 0.332 ± 0.032 and 0.012 ± 0.001 mg/mL for caffeic, gallic and chlorogenic acids, respectively, stating gallic and chlorogenic acids as the main responsible for Mac antiobesity activity. Chlorogenic acid presented no significant differences with Orlistat (*p* > 0.05), followed by gallic acid, with no significant differences with Me (*p* > 0.05), and by quercetin with an IC_50_ value of 1.105 ± 0.065 mg/mL (data not shown). In accordance with the present work, quercetin (25 μg/mL) has already been reported for inhibiting porcine pancreatic lipase by a 27.4% [39]. Caffeic acid presented the lowest inhibition capacity of the samples tested in the current work (*p* < 0.05).

### 3.5. Cell Studies

Results of cell membrane integrity through LDH-assay after 3 h of exposure time to Me extract (Figure 3A) showed a significant increase in LDH activity up to a maximum being of Triton level (positive control for cell membrane disruption), for 0.35 and 0.71 mg/mL of final extract in cell deposit media, which corresponds to 0.104 and 0.208 µM/mL quercetin in Me, respectively. From 0.71 to 2.83 mg/mL of final Me extract in cell deposit media, LDH activity decreased down to DMEM value (negative control) which could be explained by a solubility loss of Me constituents (initially soluble in 80% Ethanol) at the highest concentrations in the cell deposit medium (i.e., in DMEM) during incubation. Indeed, it was checked by absorbance measurement at 370 nm of Me deposit mixtures, as carried out before and after centrifugation (1200 rpm for 4 min, 30 °C), a decrease in absorbance by 13.6%, 21.4% and 39.4%, for the 0.71, 1.42 and 2.83 mg/mL extract concentrations, respectively, due to some precipitate formation. Such precipitate could correspond to the most hydrophobic compounds and/or marcela fibers contained in Me extract (initially soluble in 80% ethanol, but no more in DMEM, i.e., an aqueous dispersion of amino-acids, vitamins, salts and glucose). Such precipitate on cell monolayers could limit the access of harmful compounds to the cell membrane, and therefore membrane damage. We have checked that the deposit of Me extract at the highest concentrations after centrifugation to exclude insoluble compounds displayed similar LDH activity values (data not shown) than that obtained without centrifugation (Figure 3). Such precipitate was not observed for Mac extract at the studied concentrations.

For Mac extract, values of LDH activity after 3 h incubation at the tested concentrations (0.177–1.42 mg/mL in cell deposit media) were maintained below 25% of Triton^®^ value indicating no noticeable cell-membrane damage compared to control samples (DMEM ± water or ethanol) (Figure 3A).

For Me HP sample, LDH activity increased with the extract concentration as for the non-HP processed sample but significantly less steeply, presenting a maximum at 0.71 mg/mL, close to Triton level, as for the non-HP processed Me. In contrast, Mac HP sample maintained cell membrane integrity at all tested concentrations (0.177–1.42 mg/mL in cell deposit media) such as quercetin solutions (0.026–103 µM/mL), and close to DMEM level (negative control). A lower extraction of polyphenols during HP aqueous extraction might be the reason for the maintenance of cell membrane integrity, supported by total polyphenol content and bioactivity results shown above.

Generally speaking, LDH activity was higher after 22 h than 3 h of exposure time to all extract samples, and especially in the case of Me and Me HP samples (Figure 3A,B). Me induced a marked increase in LDH activity being of Triton level from lower concentrations (0.088–0.35 mg/mL in the cell deposit medium) than after 3 h incubation, followed by a significant decrease in LDH activity at ≥1.42 mg/mL extract. As previously explained for 3 h incubation, a decrease in Me constituent solubility at the highest studied concentrations (1.42–2.83 mg/mL) in cell deposit media probably explained the observed decrease in LDH activity. Such decrease in LDH activity was associated to a positive level of cell metabolic activity as evaluated by MTT-assay (Figure 3C,D).

For Mac extract, LDH activity after 22 h incubation was ≥ to that observed after 3 h, remaining ≤31% of Triton level and showing no noticeable or little cell membrane damage. For Me HP, the cell membrane integrity loss was significantly higher than after 3 h incubation and close to that observed for the non-HP processed sample. Mac HP and quercetin standard showed low LDH activity, ≤29% of Triton level indicating no or little cell membrane damage.

As for cell metabolic activity, MTT-assay after 3 h of exposure time (Figure 3C) showed significant increases in cell viability for Me and Mac samples comparing with control samples (DMEM ± water or ethanol), indicating some benefit effect of both extracts on TC7-cells. For Me extract, a high metabolic activity was maintained with increasing concentration with a maximum at 0.177 mg/mL extract in the cell deposit medium, corresponding to 0.052 µM/mL quercetin. For Mac extract, cell viability progressively increased with the extract concentration in cell deposit media reaching a maximal value for a higher concentration (1.42 mg/mL) compared to Me (0.177 mg/mL), suggesting different metabolic mechanisms for both extracts due to their composition. In contrast, purified quercetin deposed at 0.026 to 0.103 µM/mL (i.e., 0.0078 to 0.031 mg/mL) in apical cell media did not induced some significant improvement in cell viability compared to control samples (DMEM ± water or ethanol), and remained well below Me extract deposits containing similar quercetin amounts (0.088–0.354 mg/mL Me extract with 0.026–0.104 µM/mL quercetin). Me and Me HP extracts displayed similar MTT-profiles as a function of extract concentration, indicating no particular benefit or detrimental effect from HP-process. In the opposite, Mac HP presented the same tendency as Mac but with lower values remaining at the quercetin or DMEM level, possibly due to lower total polyphenol content (Table 1).

Higher increases in cell metabolic activity was observed (Figure 3D) after 22 h than 3 h of exposure to Me extract at its lowest studied concentrations (0.044–0.088 mg/mL in cell deposit media), but not at higher concentrations (namely 0.177–0.71 mg/mL) probably due to an excessive cell membrane damage as assessed by cellular LDH-leakage (Figure 3B). Figure 3D shows similar viability profiles for Me and Me HP samples after 22 h than 3 h of exposure, suggesting no detrimental effect from HP-process on ethanolic extraction. Higher increase in cell metabolic activity was observed after 22 h than 3 h of incubation with Mac extract in the range 0.354–1.42 mg/mL in apical cell media, due to the longer exposure time without noticeable cell membrane damage. In the case of Mac HP, the longer incubation time (22 h) did not improve the observed cell metabolic activity comparing with DMEM controls, possibly due to low polyphenolic content shown in Table 1.

Taking into account the whole results, the ethanolic extract displayed higher effects on TC7-cells than the aqueous extract at similar tested concentrations, with a dose-time dependence coupling mechanisms both inside the cells after constituent uptake (i.e., cell viability) and at the cell membrane surface (i.e., membrane integrity). It would be interesting to identify both kinds of active constituents.

Me and Me HP appears beneficial to TC7-cell metabolic activity at the lowest tested concentrations (0.044–0.177 mg/mL extract) and exposure time (3 h). However, higher concentrations (0.35–0.71 mg/mL extract) and exposure time (22 h) induced dramatic LDH-leakage. Indeed, a loss of cell-membrane integrity leads to further cell dysfunction then death. In contrast, Mac extract induced increased cell metabolic activity with increasing extract concentrations and exposure time, without noticeable loss of membrane integrity. However, the HP process led to a significant loss of its beneficial bioactive properties.

The fact that Me induced high levels of membrane damage and, simultaneously a high metabolic activity could result from a lag time between both mechanisms: membrane damage and decrease in mitochondrial activity; mitochondrial activity was still operating while the membrane started to be significantly damaged.

Purified quercetin deposed on TC7-cells at 0.026 to 0.103 µM/mL did not significantly increase cell metabolic activity as evaluated by MTT-assay, which does not highlight some prominent role of aglycone quercetin.

Previous reports of Polydoro et al. [25] showed ethanolic extracts (40 and 80% of ethanol) cytotoxicity assessed on Sertoli cells from Wistar rats at a concentration of 0.125 mg/mL with less than 80% of cell viability. Quercetin showed cytotoxicity [25] at high concentration (0.25 mg/mL or 0.827 µM/mL) which is higher than in the current study (0.026–0.103 µM/mL). Hence, the cytotoxicity observed by cell membrane disruption (LDH-assay) induced by ethanolic extracts in the present study might be displayed by other bioactive compounds than aglycone quercetin.

The RP-UHPLC analysis of TC7-cell methanolic extracts (Section 2.3.8) was carried out to detect the possible compound uptake by TC7-cells through their exposure to Me or Mac extracts compared to control DMEM. What was shown by RP-UHPLC chromatograms at 290 nm and/or 370 nm (Figure 4A–J) is outlined below.

A marked peak clearly appeared at 5.3 min on chromatograms at 290 nm, for most cell-extract samples after 3 h or 22 h incubation, included DMEM control samples. The latter peak detected at 290 nm but not at 370 nm, and that absorb in the UV 200–295 nm range with a maximum at 285 nm (Figure 4E), could correspond to aromatic aminoacids/peptides and protein material present in the living cells. A set of 5 to 6 ‘intermediate peaks’ that could be also interpreted as cellular metabolites were detected in the 11.6–17.4 min range at 290 nm but not at 370 nm (Figure 4E,F,I,J) for some cell-extract samples included the DMEM control sample after 22 h. Consequently, looking at the chromatograms at 370 nm that exclude the latter peaks highlights the possible presence of flavonoids that absorb at 370 nm.

Chromatograms of Mac TC7-cell-methanolic extracts at 370 nm (Figure 4C,D) did not show the typical peaks corresponding to standards of gallic, chlorogenic and caffeic acids (visible at 290 nm, Figure 1C and 370 nm Figure 4B) which presented retention times of 1.1 min, 4.6 min, 5.4 min, respectively, nor the set of specific peaks visible in the crude aqueous extract in the 12‒16.4 min RT range (Figure 4B).

In the opposite, the 3 peaks of Quercetin (18.0 min), Luteolin (18.3 min) and 3-*O*-methylquercetin (19.4 min) detected in Me crude extract at 290 nm (not shown) and more strongly at 370 nm (Figure 1D), were revealed in Me and Me HP cell-methanolic extracts analyzed at 370 nm comparing with the flat DMEM chromatogram (Figure 4A): indeed, traces for quercetin and luteolin at 18.1–18.4 min, plus a clear emerging peak at 19.4 min attributed to 3-*O*-methylquercetin were visible on Figure 4F–H,J. The lower ratio of quercetin to 3-*O*-methylquercetin found in Me TC7-cell-methanolic extracts comparing with the crude Me (Figure 1D) suggested that quercetin was poorly absorbed into TC7-cells, or further degraded/excreted in cell supernatants. Such a result was in accordance with the absence of visible peak on chromatograms at 370 nm, in the case of cell exposure to purified aglycone quercetin for 3 h or 22 h (not shown). It has been demonstrated that methylated flavonoids are better absorbed into Caco-2 cells and present a higher resistance to microsomal oxidation than their corresponding non-methylated aglycone forms [40,41] which could simply explain the present results. As indicative of the compound uptake by TC7-cells shown in Figure 4, and as estimated on the basis of chromatogram peak areas, the retained quercetin represented 0.44 ± 0.08%, 0.24 ± 0.01% and 0.16 ± 0.03% of the quercetin present in Me deposits in the case of Figure 4F,H,J, respectively. Similarly estimated, the uptake of 3-*O*-methylquecetin represented 6.86 ± 0.72%, 1.91 ± 0.15% and 4.91 ± 0.43% of the 3-*O*-methylquecetin contained in Me deposits in the case of Figure 4F,H,J, respectively. Small peaks at 22.5 and 23.5 min were also noticed after 3 h incubation (Figure 4F,G,J) suggesting the presence of newly formed derivatives at higher RT values (Figure 1B,D). More experiments are needed to achieve a quantitative evaluation of cellular uptake and thorough identification of the retained molecules.

By comparison, Mac and Mac HP samples, although deposed onto TC7-cells at higher extract concentrations (0.71–1.42 mg/mL) than Me and Me HP samples (0.088–0.354 mg/mL), revealed no peak or non-quantifiable traces on chromatograms at 370 nm in the RT range characteristic of flavonols (Figure 4C,D). These findings were in accordance with UV-Vis spectra of marcela crude extracts and TC7-cell methanolic extracts (Figure 5A,B). Indeed, while a main band (band 1) characteristic of flavonols [6,24,42] was observed at 350–365 nm in both crude Me (Figure 5A) and Me TC7-cell-methanolic extracts (Figure 5B), such a band was not found in crude Mac and Mac TC7-cell-methanolic extracts. This agrees with the fact that Mac contains low amounts of quercetin (2 mg/g aqueous extract) compared to Me (89 mg/g ethanolic extract), as well as much lower amounts of 3-*O*-methylquercetin (Figure 1). Mac TC7-cell-methanolic extracts displayed high UV absorption at 220 nm plus a broad peak with a maximum at 260–263 nm (Figure 5B). Such UV-bands also present for control DMEM could correspond to cellular material (hydrophobic amino acids/nucleic acids) solubilized by methanol during the cell-extraction step. The higher UV-light absorption in the 220–280 nm range observed for Mac and Me TC7-cell-methanolic extracts compared to control DMEM (Figure 5B) may be an indicative of enhanced TC7-cell metabolic activity induced by marcela extracts as demonstrated by MTT-assay (Figure 3C).

It is known that quercetin has highly variable and poor bioavailability, still quercetin aglycone intestinal absorption in Caco-2 cells occurs by passive diffusion and organic anion transporting polypeptide. In contrast, glycosylated form of quercetin are deglycosylated at the small intestine prior to absorption followed by quercetin metabolization through Phase II conjugation at the small intestine involving methylation, glucuronidation, and sulfation. Moreover, quercetin glucoside has been found to possess greater bioavailability due to the presence of the glucoside moiety when compared to quercetin aglycone [43].

Small intestine cell permeability for quercetin and luteolin has already been reported [40], making marcela extracts, mainly ethanolic extracts, rich in potentially bioavailable compounds. In addition, methylated flavones that show improved transport through biological membranes and increased metabolic stability compared to unmethylated flavones could present a greater oral bioavailability [40,41].

Still, bioaccessibility studies should be assessed in order to determine the stability/bioactivity of marcela bioactive compounds after digestion conditions and whether quercetin is still present in the bioaccessible fraction to be absorbed in the small intestine. Should this be the case, it would be reasonable to encapsulate the bioactive compounds into delivery systems such as liposomes or food emulsions to protect them from the extreme conditions of the gastro-intestinal tract and assess both delivery efficiency and cytotoxicity.

The incorporation of herbal extracts into traditional foods such as yogurts, cookies and meat sausages has been previously studied [44,45,46,47]; evidence suggests that food matrix and processing conditions must also be taken into account as factors that may influence bioaccessibility of the polyphenol compounds [48].

## 4. Conclusions

*Achyrocline satureioides* aqueous and ethanolic extracts presented different polyphenolic composition being characterized by phenolic acids and flavonoids, respectively. The extracts presented high polyphenol content and great antioxidant capacity determined by ABTS and ORAC-FL when compared to other medicinal plants, as well as antidiabetic (α-amylase, α-glucosidase and AGEs formation inhibition capacity) and antiobesity (pancreatic lipase inhibition capacity) activities. High hydrostatic pressure applied in the experimented conditions of pressure, pressurization duration and temperature did not prove to enhance marcela bioactive compounds extraction. Moreover, high hydrostatic pressure resulted in negative effects on some marcela bioactive properties. TC7-cell studies showed different tendencies for aqueous and ethanolic extracts as determined by LDH and MTT-assays, finding no cytotoxicity for Mac extracts at the tested concentrations (0.177–1.42 mg/mL of extract in apical cell media) compared to conventional ethanolic extracts that presented increased cell membrane disruption with increasing extract concentration. However, the lowest tested Me concentrations (0.044–0.177 mg/mL of extract in apical cell media) allowed high TC7-cell metabolic activity with limited cellular membrane damage. Cellular uptake studies revealed the presence of mainly 3-*O*-methylquercetin in Me and Me HP TC7-cell-methanolic extracts analyzed by RP-UHPLC at 370 nm, demonstrating the uptake of marcela bioactive flavonoids (mainly flavonols) into intestinal cell monolayers, in the particular case of ethanolic extracts. This suggests that marcela extracts present great potential as functional food ingredients for the prevention and/or treatment of chronic diseases.

## Figures and Tables

**Figure 1 foods-10-00893-f001:**
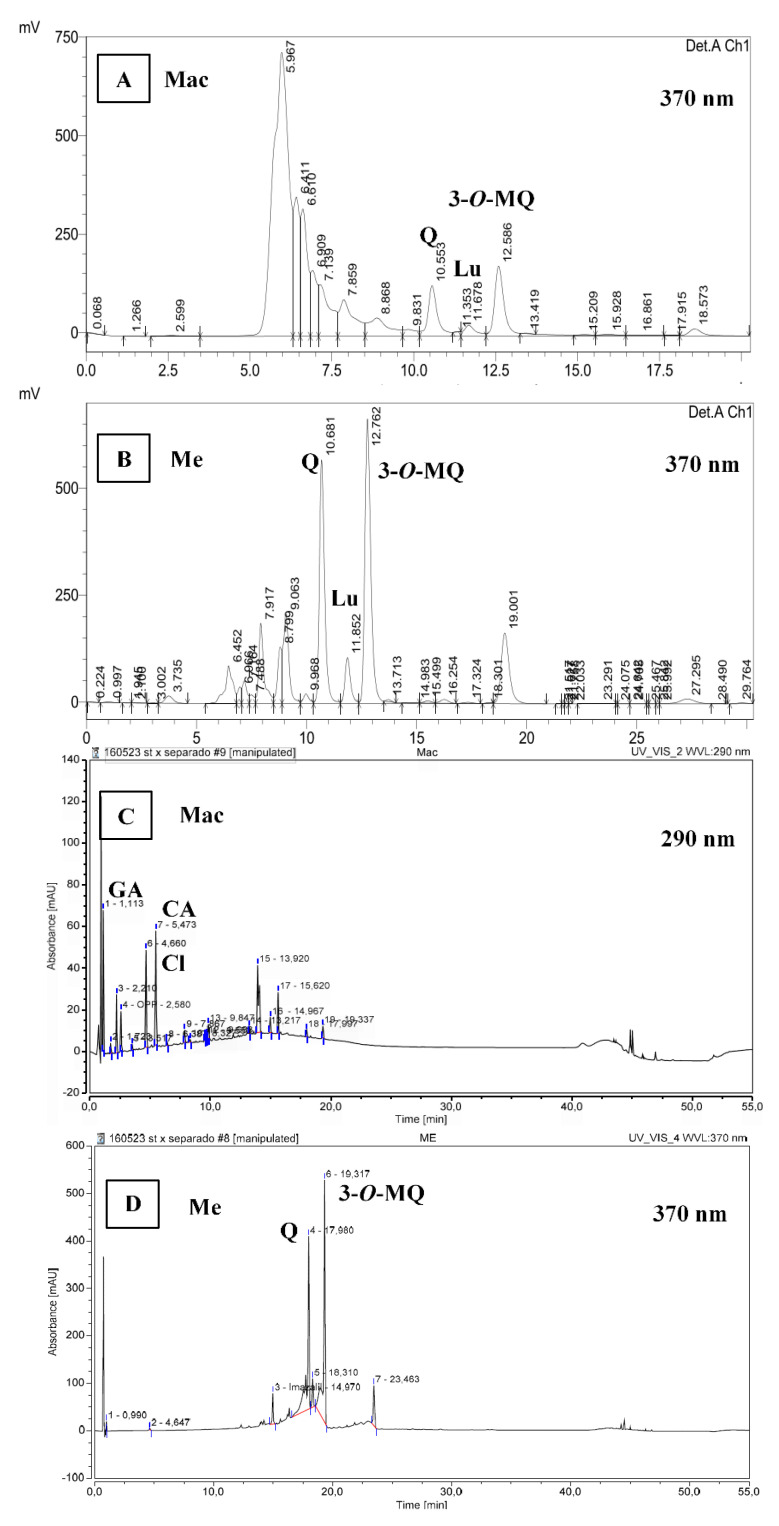
RP-HPLC chromatogram at 370 nm of samples aqueous extract (Mac 10 mg/mL) (**A**) and ethanolic extract (Me 1 mg/mL) (**B**). RP-UHPLC chromatogram at 290 nm of aqueous extract (Mac 2 mg/mL) (**C**) and at 370 nm of ethanolic extract (Me 1 mg/mL) (**D**). In order of appearance according to the retention times: GA, gallic acid; Cl, chlorogenic acid; CA, caffeic acid; Q, quercetin; Lu, luteolin; 3-*O*-MQ, 3-*O*-methylquercetin.

**Figure 2 foods-10-00893-f002:**
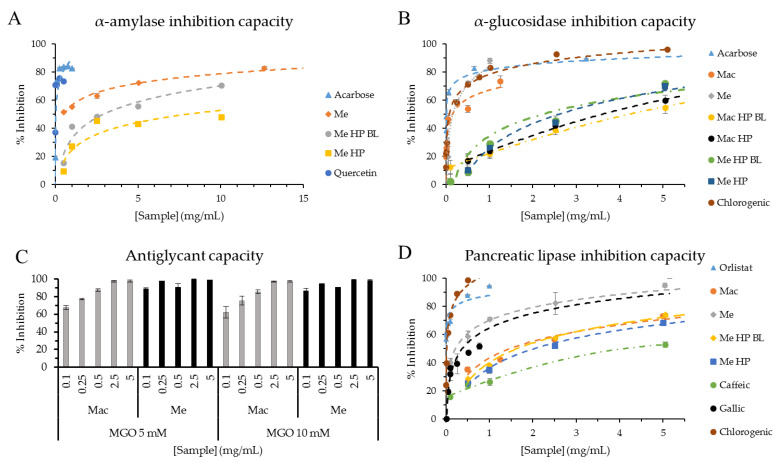
(**A**,**B**) Dose-response curves of α-amylase (**A**) and α-glucosidase (**B**) inhibition capacities expressed as % inhibition vs. sample concentration (mg/mL). (**C**) Inhibition capacity (%) of fluorescent AGEs formation with methylglyoxal (MGO) at 5 or 10 mM by different sample concentrations (mg/mL). (**D**) Dose-response curves of pancreatic lipase inhibition capacity expressed as % inhibition vs. sample concentration (mg/mL). Samples are: marcela aqueous (Mac) and ethanolic (Me) extracts, marcela aqueous HP (Mac HP) and ethanolic HP (Me HP) extracts. Blank of marcela aqueous HP (Mac HP BL) and ethanolic HP (Me HP BL) extracts. Acarbose was used as α-amylase and α-glucosidase inhibitory agent (**A**,**B**). Orlistat was used as lipase inhibitory agent (**D**). Quercetin, caffeic acid, gallic acid and chlorogenic acid were used as standards (**A**,**B**,**D**).

**Figure 3 foods-10-00893-f003:**
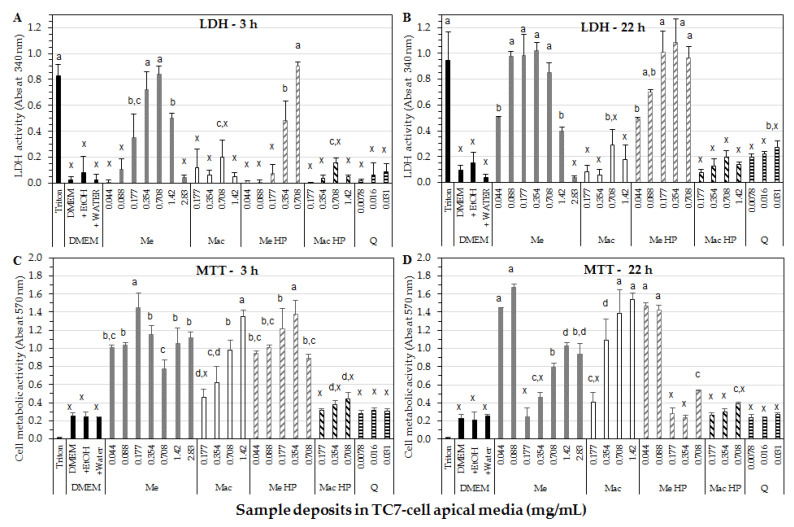
TC7-cell membrane integrity determined by LDH activity after 3 h (**A**) or 22 h (**B**) incubation, and TC7-cell viability assessed by MTT-assay after 3 h (**C**) or 22 h (**D**) incubation. Incubation of TC7-cells in the presence of marcela ethanolic extract (Me) or marcela aqueous extract (Mac) with or without HP-processing (HP), or purified quercetin (Q). Concentrations are expressed in mg/mL of marcela extracts or purified quercetin in the apical cell medium. Dulbecco’s modified Eagle medium (DMEM) (±water or ethanol) was used as negative control and Triton as positive control. Bars and error bars represent the mean values and standard deviation, respectively. For each figure, the different letters on bars state significant differences for *p* < 0.05.

**Figure 4 foods-10-00893-f004:**
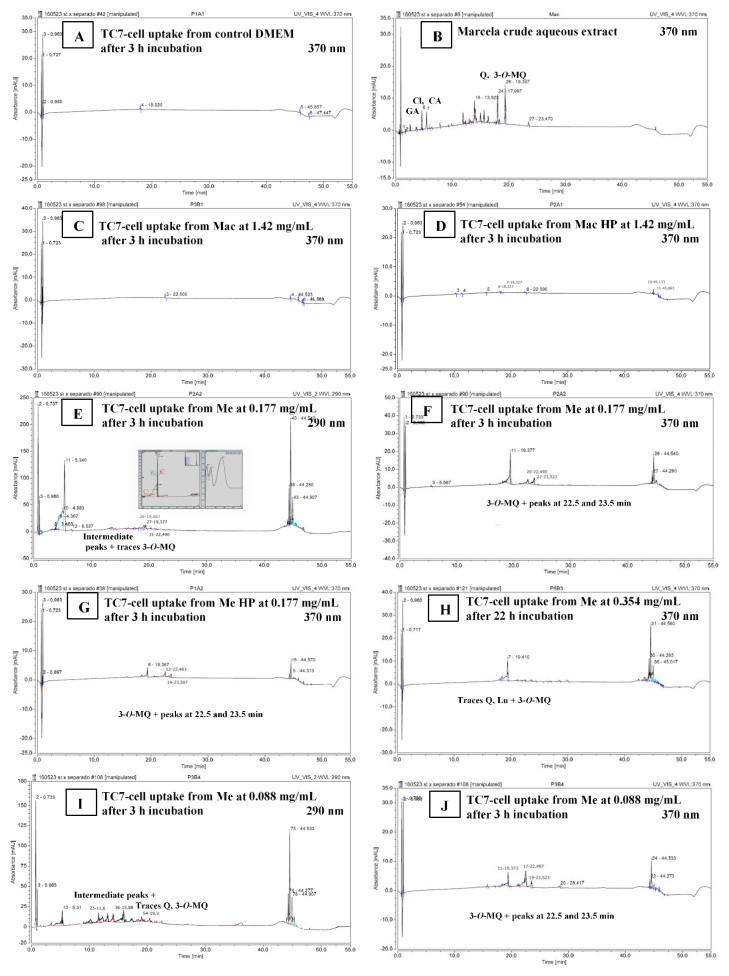
TC7-Cell uptake of marcela aqueous (Mac) and ethanolic (Me) extracts’ compounds after 3 or 22 h of incubation. (**A**) TC7-cell methanolic extract (Section 2.3.8) for control DMEM. (**B**) Crude marcela aqueous extract (2 mg/mL) given for comparison. (**C**–**J**) TC7-cell methanolic extracts (Section 2.3.8) for Mac 1.42 mg/mL (**C**), Mac HP 1.42 mg/mL (**D**), Me 0.177 mg/mL and UV 200–295 nm spectrum for the 5.3 min peak allegedly amino-acids/protein material (**E**), Me 0.177 mg/mL (**F**), Me HP 0.177 mg/mL (**G**), Me 0.354 mg/mL (**H**), Me 0.088 mg/mL (**I**,**J**). Mac and Me concentrations in TC7-Cell apical media, incubation times, and elution wavelength are indicated. Figures can be amplified on the screen.

**Figure 5 foods-10-00893-f005:**
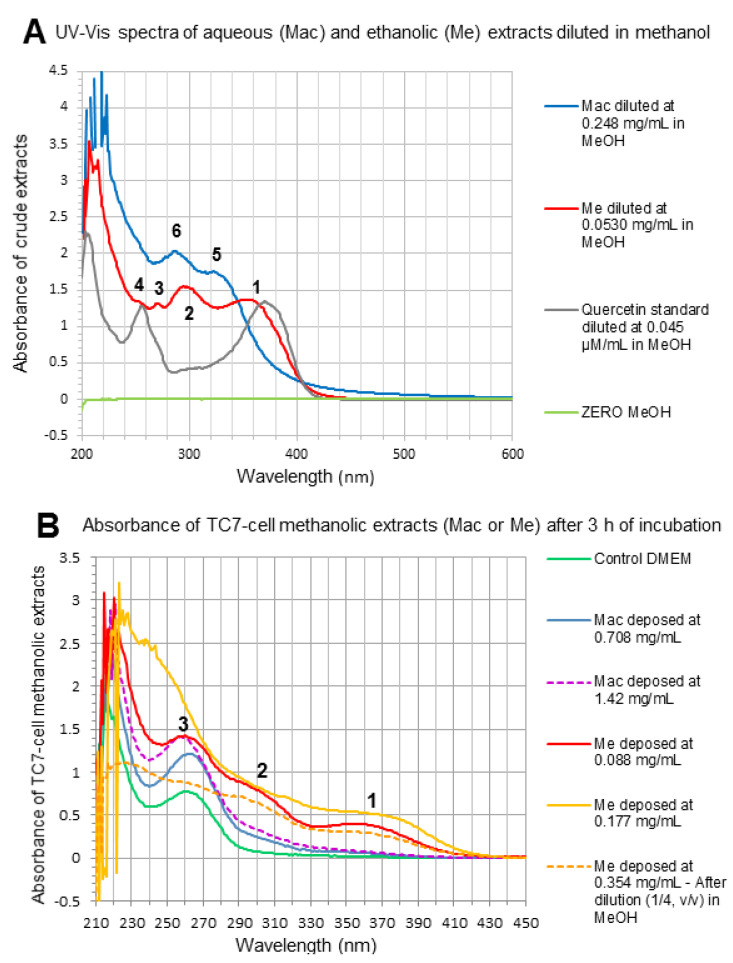
(**A**) UV-Vis spectra of crude aqueous extract (Mac), crude ethanolic extract (Me) and purified aglycone quercetin from Sigma, diluted in methanol at the indicated concentrations for absorbance measurement. Absorption maxima characteristic of: (1) B-ring absorption (band I) of flavonols (glycosylated Q, 3-*O*-MQ) and flavones (Lu); (2) hydroxycinnamic acid shoulder, flavanones, all phenolic compounds; (3) shoulder for most flavonols and flavones; (4) A-ring absorption (Band II) of flavonols, flavones; (5) hydroxycinnamic acids; (6) hydroxybenzoic acids and flavanols. (**B**) UV-Vis spectra of TC7-cell methanolic extracts after 3 h incubation with control DMEM, Mac or Me mixtures deposed at the indicated concentrations in TC7-cell apical media.

**Table 1 foods-10-00893-t001:** Results of total polyphenol content and antioxidant capacity by 2,20-azinobis-(3-ethylbenzothiazoline-6-sulfonic acid) diammonium salt (ABTS) and oxygen radical antioxidant capacity-fluorescein (ORAC-FL) methods of aqueous (Mac, Mac HP BL and Mac HP) and ethanolic (Me, Me HP BL and Me HP) extracts.

Samples	Total Polyphenol Content (mg GAE/g Extract)	ABTS (µmol TE/mg Extract)	ORAC-FL (µmol TE/mg Extract)
Mac	83.36 ± 6.69 ^c^	2.71 ± 0.27 ^a^	2.17 ± 0.10 ^d^
Me	108.79 ± 16.47 ^d^	4.19 ± 0.43 ^b^	2.30 ± 0.07 ^e^
Mac HP BL	44.07 ± 1.97 ^a^	2.06 ± 0.17 ^a^	0.34 ± 0.08 ^a^
Mac HP	42.68 ± 3.55 ^a^	2.30 ± 0.18 ^a^	0.54 ± 0.04 ^b^
Me HP BL	63.00 ± 3.12 ^b^	1.84 ± 0.20 ^a^	1.11 ± 0.08 ^c^
Me HP	58.23 ± 4.51 ^b^	1.88 ± 0.13 ^a^	1.08 ± 0.07 ^c^

Results are expressed as mean values ± SD (*n* = 3). ANOVA analysis was performed by column using Tukey test to state significant differences. Different letters indicate significant differences (*p* < 0.05) between values in the same column. Sample solutions were prepared in triplicate and assayed in triplicate. Marcela aqueous (Mac) and marcela ethanolic (Me) extracts. Marcela aqueous high pressure-assisted (Mac HP) and marcela ethanolic high pressure-assisted (Me HP) extracts. Blank of marcela aqueous HP (Mac HP BL) and marcela ethanolic HP (Me HP BL) extracts (Section 2.3.1).

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
