# Peer review of "Antioxidant, Antidiabetic, and Antiobesity Properties, TC7-Cell Cytotoxicity and Uptake of *Achyrocline satureioides* (Marcela) Conventional and High Pressure-Assisted Extracts"

_foods, 2021, doi:10.3390/foods10040893_

Round 1
Reviewer 1 Report
Overall the manuscript is well written (but I would advise that the introduction be proofread by a fluent speaker in order to improve text quality) and the topic is interesting. However, I have some concerns. The foremost being the lack of a detailed compositional profile (detailed compound profile and quantification of the identified peaks). When dealing with plant extracts, fluctuations in composition are very common which means that extract’s composition, and potentially their effect, is subject to fluctuations. Profile is then paramount when seeking to establish comparisons with literature. So please, add this information to the manuscript.
Another question I have regarding this manuscript stems from the use of a transwell assay but the lack of data reporting on the compounds present in either the apical or basolateral side of
Below a list of additional questions may be found.
Line 116: Please add the pressure used in the rotary evaporator
Line 147: Please identify which methods were used
Line 167: Please state how the compounds were identified. Was it through an online library? Comparison against standard solutions?
Line 200: What is ‘sufficient volume’?
Cell growth and seeding (section 2.3.6): Please add information regarding mycoplasma contamination screening practices.
Line 227: Authors state that TEER measurement was used but do not identify the criteria of acceptability used. Please identify.
Cell growth and seeding (section 2.3.6): Looking at the description It looks like the authors are describing the seeding in transwell systems but they do not identify the type of insert. Please provide data on membrane composition and porosity.
Sections 2.3.6 to 2.3.8: The way the information is structured is somewhat confusing. Please try to clarify it. I would prosose describing the seeding conditions for each of the assays in each of the specific sections and the addition of a new section describing the preparation of extracts for cell culture analysis.
Cell growth and seeding (section 2.3.6): No information is given on how the extracts are sterilized before being used in the assay.
Section 2.3.7. Please explain the rationale behind removing the media with extract before measuting the metabolic activity with MTT viable die. According to ISO 10993-5 for cytotoxic evaluation of medical devices, this measurement should be executed while the cells are exposed to the different stimuli.
Section 2.3.9. Please state the tests used to determine the applicability of ANOVA.
Section 3.1. Figure 1. Data regarding HP extracts is missing from the figure.
Section 3.1. Please add information regarding the composition of HP extracts.
Line 383: Please replace 403.47 with 403.5.
Line 397-8: “Quercetin seems to be the main responsible for ethanolic extracts inhibition capacity”. Why? Please clarify how you reach this conclusion.
Line 424-423: Why could this difference be? Do the authors have any hypothesis?
Line 438-440: The authors state that their data is in accordance with literature because there is a strong correlation between total phenol content, antioxidant capacity and anti-glycant capacity reported for several herbs and its the same when comparing their extracts. Was a correlation analysis carried out for the data? If so please provide the correlation coeficiens, p-values and identify the method in the materials and methods section.
Figure 2: It is hard to identify which data is which. Please try using different symbols to help differentiate data.
Figure 2 A and C: Do these graphs have a standard deviation?
Lines 480-494: If the extracts deposit it is likely that the assay is being ran at a concentration above compound solubility. In that case, should the activity observed be constant? Another possibility is that the compounds are interacting with media components, precipitating them, and possibly limiting nutrient availability but should worsen, not attenuate, membrane damage. As your extracts inhibit other enzymes is it not possible that, at higher concentrations they are directly inhibiting LDH and therefore introducing a bias into your measurement? I.e. the enzyme is there but it is not detectable with the assay used…. Was any control of enzyme activity executed in the presence of extracts? If so, please add the data.
Lines 522 – 538: Cells have high levels of membrane damage and, simultaneously a high metabolic activity? Do you have any explanation as to why this is happening?
Line 558-559: “higher bioactive effects” please specify.
Figure 4. The figure is blurred and hard to read. Please try to provide a higher quality version
Lines 621-623: Please give an idea of the concentration magnitude, even if it just a comparison of areas. Moreover, while I understand the variability associated with cell bases assays, in the materials and methods sections the authors state that they have ran triplicates for all analysis. Therefore, it should be enough to at least estimate.
Author Response
ANSWER TO REVIEWERS - Ref: foods - 1150927 - Major Revisions
First, we would like to thank the Reviewers for their contribution in the improvement of the quality of our manuscript. Changes suggested by the Reviewers have been included with *Track Changes* function in Microsoft Word on the manuscript file.
- Response to comments made by Reviewer #1:
Overall the manuscript is well written (but I would advise that the introduction be proofread by a fluent speaker in order to improve text quality) and the topic is interesting.
English has been revised in the new version of the manuscript following the suggestion made by the Reviewer.
However, I have some concerns. The foremost being the lack of a detailed compositional profile (detailed compound profile and quantification of the identified peaks). When dealing with plant extracts, fluctuations in composition are very common which means that extract’s composition, and potentially their effect, is subject to fluctuations. Profile is then paramount when seeking to establish comparisons with literature. So please, add this information to the manuscript.
We thank the Reviewer comment but unfortunately, we do not have the composition of HP-assisted extracts. Should you think this is essential, we could provide these results in 15 days’ time.
Another question I have regarding this manuscript stems from the use of a transwell assay but the lack of data reporting on the compounds present in either the apical or basolateral side of it.
We thank the Reviewer for this comment as it highlights that there is a need for further clarification. Transwell plates with ThinCert inserts were not used except for TEER measurements. For cell experiments 12-well plates (3.5 cm 2/well; 734–2156, Nunc, VWR, Fontenay-sous-Bois, France) were used as already published in the lab (Benzaria et al., 2014). Following the suggestion made by the Reviewer, the methods section has been detailed to clarify the different results obtained: “For TEER measurements, cells were grown in sterile Transwell plates with ThinCert inserts (3μm pore size; 1.13 cm2/well; Greiner Bio-one, VWR International, Fontenay-sous-Bois, France) at a density of 2.5 ´ 105 cells/well, obtaining TEER values of 750-800 Ω cm−2. Cell confluency of the cell cultures was also checked by inverse phase microscope examination.”
Below a list of additional questions may be found.
Line 116: Please add the pressure used in the rotary evaporator
The rotary evaporator condition has been added to the manuscript as “under reduced pressure, approximately 13 kPa”.
Line 147: Please identify which methods were used.
The methods have been added and described as suggested by the Reviewer. Description has been added as follows: “Briefly, protein content was determined by Kjeldhal method using the conversion factor 6.25, moisture was determined using a conventional oven at 105°C till constant weight, ashes was determined by using a furnace at 525°C for 8 hours, and fat content was obtained by Soxhlet method for 6 hours using petroleum ether.”
Line 167: Please state how the compounds were identified. Was it through an online library? Comparison against standard solutions?
We thank the Reviewer for this comment as it shows the need to clarify the HPLC methods section. We have added this phrase: “Phenolic acids and quercetin were identified and quantified by the use of pure standards and the construction of calibration curves through the detection at 290 and 370 nm for phenolic acids and quercetin, respectively.”
Line 200: What is ‘sufficient volume’?
Regarding the Reviewer question we added a detailed description of volumes as follows: “Briefly, sample mixtures consisted of 500 µL BSA stock solution (2 mg/mL in PBS, 1 mg/mL final concentration), 25/50 µL 5/10 mM MGO stock solution (200 mM in PBS, 5 mM final concentration), different volumes of extracts from a sample stock solution of 50 mg/mL (concentrations 0.1‒5 mg/mL) plus sufficient volume of PBS 10 mM pH 7.4 with 0.02% sodium azide to achieve 1 mL of the mixture final volume. Sample blanks consisted of the samples (different concentrations) with sufficient volume of PBS to achieve 1 mL of the mixture final volume (intrinsic fluorescence of the samples). Positive control was prepared by mixing 500 µL BSA, 25/50 µL 5/10 mM MGO and 475/450 µL PBS, as previously explained.”
Cell growth and seeding (section 2.3.6): Please add information regarding mycoplasma contamination screening practices.
To answer the Reviewer comment, it has been indicated in the text: “All cell seeding and sample deposit experiments were carried in sterile conditions under a laminar flow cabinet (PSM MSC Advantage, ThermoFisher Scientific, Saint-Herblain, France), using 0.2 µm filtrated media, sterile solutions and sterile plastic material (pipets, tips, flasks, plates, plates, microplates, Eppendorf ® and Falcon® tubes)”.
Line 227: Authors state that TEER measurement was used but do not identify the criteria of acceptability used. Please identify.
To answer the Reviewer comment, it has been added in the text: “For TEER measurements, cells were grown in sterile Transwell plates with ThinCert inserts (3μm pore size; 1.13 cm2/well; Greiner Bio-one, VWR International, Fontenay-sous-Bois, France) at a density of 2.5 ´ 105 cells/well, obtaining TEER values of 750-800 Ω cm−2. Cell confluency of the cultures was also checked by inverse phase microscope examination”.
Cell growth and seeding (section 2.3.6): Looking at the description It looks like the authors are describing the seeding in transwell systems but they do not identify the type of insert. Please provide data on membrane composition and porosity.
Sections 2.3.6 to 2.3.8: The way the information is structured is somewhat confusing. Please try to clarify it. I would propose describing the seeding conditions for each of the assays in each of the specific sections and the addition of a new section describing the preparation of extracts for cell culture analysis.
The seeding conditions are the same for all experiments and described in section 2.3.6. The preparation of sample deposit solutions are explained in 2.3.6. The determination of in vitro TC7-cell membrane integrity and cell metabolic activity evaluated after sample deposit are explained in 2.3.7. The determination of cellular uptake of marcela compounds in TC7-cells is explained in 2.3.8. The title of 2.3.6 section has been changed for more clarity.
Cell growth and seeding (section 2.3.6): No information is given on how the extracts are sterilized before being used in the assay.
Ethanolic solution of marcela (Me) in 95% or 80% ethanol does not need some sterilization. Marcela aqueous extract solution was obtained after 1 h boiling. The corresponding dehydrated Mac extract was added to sterile distilled water for cell deposits.
Section 2.3.7. Please explain the rationale behind removing the media with extract before measuring the metabolic activity with MTT viable die. According to ISO 10993-5 for cytotoxic evaluation of medical devices, this measurement should be executed while the cells are exposed to the different stimuli.
Section 2.3.9. Please state the tests used to determine the applicability of ANOVA.
ANOVA was applied because the samples are independent and present a normal distribution.
Section 3.1. Figure 1. Data regarding HP extracts is missing from the figure.
We thank the reviewers comment but unfortunately, we do not have the composition of HP-assisted extracts. Should you think this is essential, we could provide these results in 15 days’ time.
Section 3.1. Please add information regarding the composition of HP extracts.
We thank the Reviewers comment but unfortunately, we do not have the composition of HP-assisted extracts. Should you think this is essential, we could provide these results in 15 days’ time.
Line 383: Please replace 403.47 with 403.5.
It has been corrected in the manuscript.
Line 397-8: “Quercetin seems to be the main responsible for ethanolic extracts inhibition capacity”. Why? Please clarify how you reach this conclusion.
We thank this comment from the Reviewer as it led us to a deeper thinking of the matter. We stated the α-amylase inhibition capacity of quercetin standard (IC50 = 2.4 ± 0.2 µg/mL) that led to our thinking that quercetin could be the main responsible because of being one of the main compounds present in ethanolic extract. However, we should have taken 3-O-methylquercetin into account, which is the other main compound. Unfortunately, we did not and do not have this particular standard in order to determine its α-amylase inhibition capacity. In sum, we should have written as follows: “Moreover, quercetin seems to be one of the responsible for ethanolic extracts inhibition capacity, because of being one of the main compounds present in the latter extracts.”
Line 424-423: Why could this difference be? Do the authors have any hypothesis?
We apologize for the error but line 424-425 should have said: “As to polyphenols standards, quercetin has shown to possess more α-amylase inhibition capacity than acarbose [34] in accordance with the current results”. Nyambe-Silavwe et al. (2015) presented the results of the IC50 in µM for acarbose and quercetin standards, 3.5 and 19.8 µM respectively, but in our work are expressed as µg/mL, 34.1 and 2.4 µg/mL respectively. The equivalent of the IC50 in µg/mL would be 12.78 and 1.06, respectively. The phrase has been corrected.
Line 438-440: The authors state that their data is in accordance with literature because there is a strong correlation between total phenol content, antioxidant capacity and anti-glycant capacity reported for several herbs and it’s the same when comparing their extracts. Was a correlation analysis carried out for the data? If so please provide the correlation coefficients, p-values and identify the method in the materials and methods section.
We thank the Reviewer for this comment as it highlights the need for re-write the sentence as follows: “…showed correlation between total polyphenol content, antioxidant capacity and anti-glycant capacity [35], showing the same tendency when comparing marcela aqueous and ethanolic conventional extracts. Ethanolic extract (Me) showed higher total polyphenol content, antioxidant and anti-glycant capacity than aqueous extract (Mac)”.
Figure 2: It is hard to identify which data is which. Please try using different symbols to help differentiate data.
Figure 2 has been modified for better identification of the data.
Figure 2 A and C: Do these graphs have a standard deviation?
Standard deviations in Figure 2A are not visible because the values are very small and included in the symbols. Standard deviations in Figure 2C have been added to the graph.
Lines 480-494: If the extracts deposit it is likely that the assay is being ran at a concentration above compound solubility. In that case, should the activity observed be constant? Another possibility is that the compounds are interacting with media components, precipitating them, and possibly limiting nutrient availability but should worsen, not attenuate, membrane damage. As your extracts inhibit other enzymes is it not possible that, at higher concentrations they are directly inhibiting LDH and therefore introducing a bias into your measurement? I.e. the enzyme is there but it is not detectable with the assay used…. Was any control of enzyme activity executed in the presence of extracts? If so, please add the data.
Me extract compounds were initially soluble in 80% ethanol and became partly insoluble in DMEM when deposed at the highest concentration in cell apical media. We have checked the LDH-response after deposit on TC7-cells, of Me extracts at the highest concentrations after centrifugation (excluding the pellet of centrifugation): similar LDH-responses were obtained than without a previous centrifugation. This suggests that the most harmful compounds (the most hydrophobic?) were excluded through their precipitation. It is also possible that some hydrocolloids (Marcela contains 57% fibers w/w) initially soluble in 80% ethanol precipitated in DMEM (DMEM is an aqueous dispersion of amino-acids, vitamins, salts and glucose) making a gel layer on the cells which limit the access of harmful compounds to the cell membrane. Controls of LDH activity carried out on TC7-cell apical media after deposit of increasing volume of Triton or Me (0.354 mg/mL) solutions showed regularly increasing LDH-response up to 1-1.5 absorbance units which did not allow any other explanation.
Lines 522 – 538: Cells have high levels of membrane damage and, simultaneously a high metabolic activity? Do you have any explanation as to why this is happening?
It is possible that there is a lag time between both mechanisms: membrane damage and decrease in mitochondrial activity: mitochondrial activity is still operating while the membrane start to be significantly damaged. The remark has been added in the text.
Line 558-559: “higher bioactive effects” please specify.
The term bioactive has been removed.
Figure 4. The figure is blurred and hard to read. Please try to provide a higher quality version
To answer the Reviewer remark, Figure 4 has been re-loaded more carefully.
Lines 621-623: Please give an idea of the concentration magnitude, even if it just a comparison of areas. Moreover, while I understand the variability associated with cell bases assays, in the materials and methods sections the authors state that they have ran triplicates for all analysis. Therefore, it should be enough to at least estimate.
It was effectively not easy to carry out cell-extraction experiments without losing cells. Not all the triplicates were completely exploitable. More experiments are needed to achieve a quantitative evaluation of cellular uptake and thorough identification of the retained molecules. Nevertheless, as suggested by the Reviewer, we have added some estimation on the basis of chromatogram peak areas, for the retained quercetin and 3-O-methylquercetin in the case of duplicate TC7-cell methanolic extracts corresponding to Figure 4F, H, and J.
I am attaching the manuscript with the correction.

Reviewer 2 Report
foods-1150927-peer-review-v1
I would like to appreciate the authors’ effort towards bringing a comprehensive analysis on the chemical composition and metabolic-related bioactivities of Achycocline satureoides, a plant used in South-American folk medicine; using high hydrostatic pressure-assisted extracts and Caco-2/TC7 human intestinal cells. The relevance of this work is double, since not only assesses the bioactivities exerted by the extracts, but also assesses alternative extraction methods. However, the authors must clarify how they applied extracts to the cell cultures, since the described method would result in ensured cytotoxicity.
The manuscript needs substantial editing to meet quality standards required for publication.
Abstract
L22: State whether if its used in traditional medicine or if “native”, where is native to.
L32: Please change “stated” for a suitable verb.
Introduction
L43: Usefulness or potential of using natural sources is not well-introduced. It is stated as a correlation with the occurring metabolic diseases. Please rephrase or elaborate to properly justify the choice of studying this medicinal plant for the proposed diseases.
L61: Albeit intuitive, it is better to state why the choice of solvent (more/less polar) alters the resulting phenolic profile.
L62: You mean different properties or different effectiveness response of these?
L67: Please rephrase this whole sentence to ease readability.
L71: Again, it is hard to see this correlation. Please better elaborate on this and write a statement reflecting on that bioactivity and cytotoxicity studies are used to assess potential effectiveness of the described compounds from this specific source. In addition, since you have made absorption experiments, it would be nice to point it out, making clear that these compounds could exert the mentioned bioactivities once they are absorbed.
Additionally, introduce and clearly state how and why pure quercetin is being used in this work.
Materials and Methods
Please, put the characteristics of Caco-2/TC7 cells in the subsection of Cell culture, in the Materials section. Differentiate between the proper cells and all the chemicals needed for the culture medium, as it is described.
Please indicate the nature of the pancreatic lipase and what AGES refers to.
You must clearly indicate how and why phenolic acids (as well as quercetin) have been used like this in this work.
On the use of extracts abbreviations, I think that using “Me” for ethanolic extract can be misleading, and be mistaken as methanol. As the aqueous extract is labeled as “Mac”, perhaps the M refers to “marcela”. Nevertheless, it would be best to make a change of labels in order to not hinder interpretation.
L147: Please, summarize or briefly describe these methods.
L229: Please clarify how and why the times of exposure were applied and in how many cell culture samples.
L231-239: Are you meaning that you did solubilize aqueous extracts in distilled water and ethanolic extracts in 80% ethanol, then applied them to the cell cultures? The issue is that almost a third of the solubilized solution is the named solvent. This could arguably bias all your results related to in vitro bioactivity assays.
Results
Figure 1: Please provide an enhanced quality version of these chromatograms.
L408-435: This whole discussion doesn’t add meaningful information. Keep in mind that this work has been submitted to the Foods journal. Citing other medicinal plants or comparable results from these doesn’t stablish a correlation. As such, not only both these paragraphs must be completely rewritten using suitable references, but you must properly discuss these results under the scope of the journal.
L474-477: Again, this is not adding any meaningful information or context to your results. Please change.
Figure 4: As with figure 1, please provide figures with enhanced quality. Additionally, these are almost unreadable so it would be best to amplify their size. Moreover, why is there another graph in E?
L642: Rephrase this statement, as they invalidate each other.
Final Remarks
While the work has points of relevance and potential, the described issues; especially on cell culture methodlogy, must be addressed and extensive editing is required.
Author Response
First, we would like to thank the Reviewers for their contribution in the improvement of the quality of our manuscript. Changes suggested by the Reviewers have been included with *Track Changes* function in Microsoft Word on the manuscript file. I am attaching the manuscrip with the corrections
- Response to comments made by Reviewer #2:
I would like to appreciate the authors’ effort towards bringing a comprehensive analysis on the chemical composition and metabolic-related bioactivities of Achycocline satureoides, a plant used in South-American folk medicine; using high hydrostatic pressure-assisted extracts and Caco-2/TC7 human intestinal cells. The relevance of this work is double, since not only assesses the bioactivities exerted by the extracts, but also assesses alternative extraction methods. However, the authors must clarify how they applied extracts to the cell cultures, since the described method would result in ensured cytotoxicity.
Following the suggestion made by the reviewer, we clarified how we applied the extracts to the cell cultures in the methods section.
The manuscript needs substantial editing to meet quality standards required for publication.
The manuscript has been carefully revised and edited. We hope now we meet the quality standards.
Abstract
L22: State whether if its used in traditional medicine or if “native”, where is native to.
The sentence has been rewritten.
L32: Please change “stated” for a suitable verb.
The verb “stated” has been changed for “highlighted”.
Introduction
L43: Usefulness or potential of using natural sources is not well-introduced. It is stated as a correlation with the occurring metabolic diseases. Please rephrase or elaborate to properly justify the choice of studying this medicinal plant for the proposed diseases.
The order of the Introduction has been modified according to the suggestion of the Reviewer. If it is not enough, please be more specific.
L61: Albeit intuitive, it is better to state why the choice of solvent (more/less polar) alters the resulting phenolic profile.
To answer the Reviewer remark, the phrase has been rewritten as follows: “Aqueous and ethanolic extracts have shown different polyphenolic profiles as a consequence of different polyphenols solubility correspondent to solvent polarity, with subsequently different bioactive properties and/or biological effectiveness. High hydro-static pressure (HP) technology proved to increase polyphenolic extraction yields [7] and plant cell membrane damage [8]”.
L62: You mean different properties or different effectiveness response of these?
The sentence has been changed as follows: “Aqueous and ethanolic extracts have shown different polyphenolic profiles as a consequence of different polyphenols solubility correspondent to solvent polarity, with subsequently different bioactive properties and/or biological effectiveness, making …”
L67: Please rephrase this whole sentence to ease readability.
We thank the Reviewer for pointing out that the phrase is not clear. It has been rephrased.
L71: Again, it is hard to see this correlation. Please better elaborate on this and write a statement reflecting on that bioactivity and cytotoxicity studies are used to assess potential effectiveness of the described compounds from this specific source. In addition, since you have made absorption experiments, it would be nice to point it out, making clear that these compounds could exert the mentioned bioactivities once they are absorbed.
In response to the Reviewer comment, we changed the Introduction as follows: “These compounds possess several bioactive properties such as cytoprotective activity against oxidant agents [2], but there are no reports on the bioavailability and/or absorption experiments, neither about its cytotoxicity on intestinal cells (as a means to elucidate the effect after their ingestion), which are necessary to assess the potential effectiveness of the marcela bioactive compounds. Once absorbed, these compounds may exert the above mentioned bioactivities”.
Additionally, introduce and clearly state how and why pure quercetin is being used in this work.
A description in “Materials and Methods” section has been added for phenolic acids and quercetin standards use.
Materials and Methods
Please, put the characteristics of Caco-2/TC7 cells in the subsection of Cell culture, in the Materials section. Differentiate between the proper cells and all the chemicals needed for the culture medium, as it is described.
The title of sections 2.1 and 2.2 have revised for more clarity.
Please indicate the nature of the pancreatic lipase and what AGES refers to.
Pancreatic lipase nature has been added to section 2.1. AGEs abbreviation stands for advanced glycation end products and has been clarified in section 2.3.5. Additionally, we have added α-amylase detailed description.
You must clearly indicate how and why phenolic acids (as well as quercetin) have been used like this in this work.
Details about the phenolic acids and quercetin standards have been added to section 2.1.: “Phenolic acids (gallic, chlorogenic and caffeic acids) and quercetin standards were purchased from Sigma-Aldrich (St. Louis, MO, USA) and used for marcela extract composition RP-HPLC and RP-UHPLC analyses”.
On the use of extracts abbreviations, I think that using “Me” for ethanolic extract can be misleading, and be mistaken as methanol. As the aqueous extract is labeled as “Mac”, perhaps the M refers to “marcela”. Nevertheless, it would be best to make a change of labels in order to not hinder interpretation.
We prefer not to change the extract abbreviations in all the text, Figures and Table, which will probably leave errors. Mac and Me have been clearly defined in section 2.3.1. When methanol is named, the term methanol is written as such, without abbreviation, to avoid misleading.
L147: Please, summarize or briefly describe these methods.
The methods have been added and described as suggested by the Reviewer. Description has been added as follows: “Briefly, protein content was determined by Kjeldhal method using the conversion factor 6.25, moisture was determined using a conventional oven at 105°C till constant weight, ashes was determined by using a furnace at 525°C for 8 hours, and fat content was obtained by Soxhlet method for 6 hours using petroleum ether.”
L229: Please clarify how and why the times of exposure were applied and in how many cell culture samples.
Times of TC7-cell exposure (3h or 22h) were chosen on the basis of previous experiments, and taking into account the open time necessary to prepare cell series (sample deposit onto TC7-cells in the 12-well plates, incubation of the cells in the incubator after deposit, recovering of apical cell media on ice for the further LDH evaluation, and treatment of cell monolayers with MTT). The cells were treated by plates (12-well plates) and incubation time or treatment time was counted independently for each 12-well plate.
The whole repetition of experiments is indicated in Section 2.3.7.
L231-239: Are you meaning that you did solubilize aqueous extracts in distilled water and ethanolic extracts in 80% ethanol, then applied them to the cell cultures? The issue is that almost a third of the solubilized solution is the named solvent. This could arguably bias all your results related to in vitro bioactivity assays.
As indicated in Section 2.3.6., a 100 μL of each extract solution (Me or Mac) was mixed with 1.9 mL of Phenol red-free DMEM (1/20 dilution) for cell deposit in the 12-well plates. A 500 µL of the latter mixtures was deposed by well on the apical side of TC7-cells. To check the possible effect of distilled water or 80% ethanol on the cells, controls were prepared for each series, mixing 100 µL of distilled water or 100 µL of 80% ethanol solution with 1.9 mL of Phenol red-free DMEM (1/20 dilution; final 4% ethanol) for cell deposit in the 12-well plates. A 500 µL of the latter mixtures was deposed by well on the apical side of TC7-cells. Comparing to DMEM alone deposed on the cells as control, the results of Figure 3 (A-D) indicated that there was no significant difference between DMEM alone, DMEM plus distilled water, or DMEM plus 80% ethanol, which excluded a possible bias in the cell experiments.
Results
Figure 1: Please provide an enhanced quality version of these chromatograms.
Figure 1 has been re-loaded more carefully.
L408-435: This whole discussion doesn’t add meaningful information. Keep in mind that this work has been submitted to the Foods journal. Citing other medicinal plants or comparable results from these doesn’t stablish a correlation. As such, not only both these paragraphs must be completely rewritten using suitable references, but you must properly discuss these results under the scope of the journal.
All the discussion has been modified according to the comment of the Reviewer.
L474-477: Again, this is not adding any meaningful information or context to your results. Please change.
To answer the Reviewer remark, we have erased the paragraph and a phrase about quercetin pancreatic lipase inhibition has been added: “In accordance with the present work, quercetin (25 μg/mL) has already been reported for inhibiting porcine pancreatic lipase by a 27.4% [36]”.
Figure 4: As with figure 1, please provide figures with enhanced quality. Additionally, these are almost unreadable so it would be best to amplify their size. Moreover, why is there another graph in E?
Figure 4 has been re-loaded more carefully.
It became easy to amplify the Figures when reading on screen. The small graph in Figure 4E shows the UV spectrum of the peak 5.3 min supposed to be amino-acids/protein material.
L642: Rephrase this statement, as they invalidate each other.
In response to the Reviewer comment, a word has been changed.
Final Remarks
While the work has points of relevance and potential, the described issues; especially on cell culture methodology, must be addressed and extensive editing is required.
Extensive editing has been done especially on cell culture methodology.

Reviewer 3 Report
The manuscript entitled "Antioxidant, antidiabetic, and antiobesity properties, intestinal cell cytotoxicity and uptake assessment of Achyrocline satureioides (‘marcela’) conventional and high hydrostatic pressure-assisted extracts" by Fernández-Fernández et al, presents interesting results about the bioactive properties od Achyrocline satureioides. Some problems must be addressed:
-The title is too long. I suggest to modify to a more concise and informative title.
-In the first paragraph of Introduction, more references should be added after the sentence "The typical metabolic disorders..."
-Why the section "2.2. Cell culture" is not presented before the procedures using the cells?
-Please uniformize: 'marcela', "Marcela", "marcela"... I suggest using only marcela.
-What are the blanks presented in Table 1 (Mac HP BL and Me HP BL)? Please explain.
-Please correct and uniformize the names of the bioactive compounds along the entire manuscript. For example, the "O" of "3-O-methylquercetin" should be in italics.
-The Conclusion is also too long. I suggest to summarize with the main conclusion of the work.
Author Response
First, we would like to thank the Reviewers for their contribution in the improvement of the quality of our manuscript. Changes suggested by the Reviewers have been included with *Track Changes* function in Microsoft Word on the manuscript file. I am attaching the manuscript with corrections.
- Response to comments made by Reviewer #3:
The manuscript entitled "Antioxidant, antidiabetic, and antiobesity properties, intestinal cell cytotoxicity and uptake assessment of Achyrocline satureioides (‘marcela’) conventional and high hydrostatic pressure-assisted extracts" by Fernández-Fernández et al, presents interesting results about the bioactive properties of Achyrocline satureioides. Some problems must be addressed:
-The title is too long. I suggest to modify to a more concise and informative title.
The title has been shortened as possible.
-In the first paragraph of Introduction, more references should be added after the sentence "The typical metabolic disorders..."
The Introduction has been modified according to the Reviewer comment and the phrase has been removed.
-Why the section "2.2. Cell culture" is not presented before the procedures using the cells?
The titles of sections 2.1 and 2.2 have been modified for more clarity.
-Please uniformize: 'marcela', "Marcela", "marcela"... I suggest using only marcela.
It has been uniformized in all the manuscript as suggested by the Reviewer.
-What are the blanks presented in Table 1 (Mac HP BL and Me HP BL)? Please explain.
The legend of Table 1 has been completed.
-Please correct and uniformize the names of the bioactive compounds along the entire manuscript. For example, the "O" of "3-O-methylquercetin" should be in italics.
The "O" of "3-O-methylquercetin" has been changed to italics in all the manuscript.
-The Conclusion is also too long. I suggest to summarize with the main conclusion of the work.
Conclusion has been shortened as possible.

Reviewer 4 Report
The paper entitled “Antioxidant, antidiabetic, and antiobesity properties, intestinal cell cytotoxicity and uptake assessment of Achyrocline satu- reioides (‘marcela’) conventional and high hydrostatic pressure- assisted extracts”. The aim of the present work is to evaluate Achyrocline satureioides antioxidant, antidi- abetic and antiobesity properties of aqueous and ethanolic extracts compared to HP-as- sisted extracts, along with the exposure to cultures of intestinal cells in order to elucidate the degree of cytotoxicity and uptake of extracted bioactive compounds.
This work is good. The entire manuscript should be corrected.
- line 55: should be 3-O-methylquercetin - where "O" should be italic. Please check out the full article and correct it.
- I didn't see the "reference" section of this article so I can not check whether the literature has been cited appropriately in the "material and methods" section. If I dont see the "reference" section authors should describe the methods used in detail.
- Table 1. Please show the results with 2 places after the dot.
Author Response
First, we would like to thank the Reviewers for their contribution in the improvement of the quality of our manuscript. Changes suggested by the Reviewers have been included with *Track Changes* function in Microsoft Word on the manuscript file. I am attaching the mansucript file.
Response to comments made by Reviewer #4:
The paper entitled “Antioxidant, antidiabetic, and antiobesity properties, intestinal cell cytotoxicity and uptake assessment of Achyrocline satureioides (‘marcela’) conventional and high hydrostatic pressure- assisted extracts”. The aim of the present work is to evaluate Achyrocline satureioides antioxidant, antidiabetic and antiobesity properties of aqueous and ethanolic extracts compared to HP-assisted extracts, along with the exposure to cultures of intestinal cells in order to elucidate the degree of cytotoxicity and uptake of extracted bioactive compounds.
This work is good. The entire manuscript should be corrected.
- line 55: should be 3-O-methylquercetin - where "O" should be italic. Please check out the full article and correct it.
The "O" of "3-O-methylquercetin" has been changed to italics in all the manuscript.
- I didn't see the "reference" section of this article so I can not check whether the literature has been cited appropriately in the "material and methods" section. If I don’t see the "reference" section authors should describe the methods used in detail.
In response to the Reviewer comment, references from the “Materials and Methods” have been double-checked, just in case.
- Table 1. Please show the results with 2 places after the dot.
The whole Table has been changed in response to the Reviewer comment. All the results have been expressed with 2 places after the dot.

Round 2
Reviewer 2 Report
I appreciate the authors’ effort towards improving the quality of the manuscript and paying close attention to the various reviewers’ comments. Yet, some issues remain to be checked:
L55: As it was corrected for the abstract, so it should be for the introduction. Please check the phrasing.
L87: Despite being more elaborated, abbreviations are without proper introduction.
L264: While further detail is appreciated, if “previous experiments” are mentioned, they should be accompanied by a reference.
L449-451: Once again, this doesn’t’ add meaningful information. Simply being traditional remedies doesn’t mean that they would be effective or share the same properties. You did, however, phrase this perfectly in L519, by relating the observed effects to quercetin content, hence to the plant composition. Check and try doing so here.
Figure 4: While this explanation could be satisfactory, you must include it in the Figure’s subtitle.
As suggested in the previous review, the manuscript should clearly relate the work to foods, as this is the scope of the journal. Some brief lines (with suitable references) describing its use as food or beverage or its potential blending with foodstuff should be enough. The only references to this in both Introduction and Conclusions are unfortunately, too brief.
The authors have extensively edited the manuscript and I sincerely think this has improved the quality of the work. I would consider the work ready for publication after minor changes.
Reviewer 3 Report
The authors adressed all my comments and corrected the manuscript accordingly.
Author Response
There are no comments to incorporate
Reviewer 4 Report
Accept in present form
Author Response
There are no comments to incorporate